# Modular in vivo assembly of *Arabidopsis* FCA oligomers into condensates competent for RNA 3′ processing

Geng-Jen Jang[1], Alex L Payne-Dwyer[2], Robert Maple [ID][1], Zhe Wu[1,3], Fuquan Liu [ID][1,4], Sergio G Lopez [ID][1], Yanning Wang [ID][5], Xiaofeng Fang [ID][5], Mark C Leake [ID][2,6] & Caroline Dean [ID][1][✉]

## Abstract

Our understanding of the functional requirements underpinning biomolecular condensation in vivo is still relatively poor. The *Arabidopsis* RNA binding protein FLOWERING CONTROL LOCUS A (FCA) is found in liquid-like nuclear condensates that function in transcription termination, promoting proximal polyadenylation at many target genes in the *Arabidopsis* genome. To further understand the properties of these condensates in vivo, we used single-particle tracking experiments on FCA reporters stably expressed at endogenous levels in plant nuclei. SEC-MALS analyses suggested that FCA forms a core oligomer consistent with a size of four molecules; in vivo particle tracking indicated that this core molecule multimerizes into higher-order particles. The ensuing assemblies coalesce into macromolecular condensates via the coiled-coil protein FLL2, which is genetically required for FCA function. Accordingly, FLL2 predominately co-localizes with FCA in larger-sized condensates. A missense mutation in the FCA RRM domain, also genetically required for FCA function, reduced average size of both FCA particles and condensates, but did not perturb the core oligomer. Our work points to a modular structure for FCA condensates, involving multimerization of core oligomers assembled into functional macromolecular condensates via associated RNA and FLL2 interactions.

**Keywords** Biomolecular Condensate; *COOLAIR*; *FLC*; Non-coding RNA; RNA Processing
**Subject Categories** Plant Biology; RNA Biology

## Introduction

Cellular compartmentalization by biomolecular condensates is associated with diverse processes in eukaryotic cells (Banani et al, 2017; Shin & Brangwynne, 2017). While in vitro studies have highlighted their roles in increasing local concentration and residence time of interacting components, the functional characteristics of heterogeneous condensates in vivo remain unclear. Plants are an excellent system in which to study biomolecular condensates due to their response to external conditions (Field et al, 2023). For example, Arabidopsis *FLOWERING LOCUS C* (*FLC*), a gene encoding a key flowering repressor, is regulated by co-transcriptional and antisense-mediated chromatin silencing mechanisms (Fang et al, 2020; Xu et al, 2021) that involve multiple nuclear biomolecular condensates with very different properties (Berry et al, 2017; Fang et al, 2019; Zhu et al, 2021; Fiedler et al, 2022). One of those involves the RNA-binding protein FCA, which forms liquid-like nuclear condensates (Fang et al, 2019). Another involves FRIGIDA, a strong activator of *FLC* expression, which forms condensates at low temperature, with turnover dynamics that are slow compared to FCA (Zhu et al, 2021). FCA directly interacts with FY, a Pfs2p/WDR33 homologue and a central component of the polyadenylation machinery in eukaryotes (Simpson et al, 2003). FCA contains two RRM (RNA Recognition Motif) domains and a WW domain located between two predicted prion-like domains (Macknight et al, 1997). FCA recruits RNA 3' processing factors to an R-loop, which is a DNA:RNA hybrid generated by the antisense non-coding RNA, *COOLAIR*, to induce transcription-coupled chromatin repression (Xu et al, 2021; Mateo-Bonmati et al, 2024; Menon et al, 2024). This mechanism also involves an FCA-interacting coiled-coil protein, FLL2, which is required for both formation of FCA nuclear condensates and FCA function (Fang et al, 2019). In vitro assays revealed that FCA condensates were enhanced by addition of low amounts of Arabidopsis RNA. However, the immunoprecipitation of *COOLAIR* nascent transcripts by FCA was not dependent on FLL2 (Fang et al, 2019).

Knowledge of the properties and molecular structure of biomolecular condensates helps dissection of the molecular mechanisms involved (Peran and Mittag, 2020). However, studies of biological condensates have focused on in vitro analysis and transfection studies and if in vivo stable expression fusions have been made, they are often not carefully matched to endogenous expression levels. This will

[1]Department of Cell and Developmental Biology, John Innes Centre, Norwich Research Park, Norwich, UK. [2]School of Physics, Engineering and Technology, University of York, York, UK. [3]Key Laboratory of Molecular Design for Plant Cell Factory of Guangdong Higher Education Institutes, Institute of Plant and Food Science, Department of Biology, School of Life Sciences, Southern University of Science and Technology, Shenzhen, China. [4]Institute of Global Food Security, School of Biological Sciences, Queen's University Belfast, Belfast, UK. [5]Center for Plant Biology, School of Life Sciences, Tsinghua University, Beijing, China. [6]Department of Biology, University of York, York, UK.
[✉]E-mail: caroline.dean@jic.ac.uk

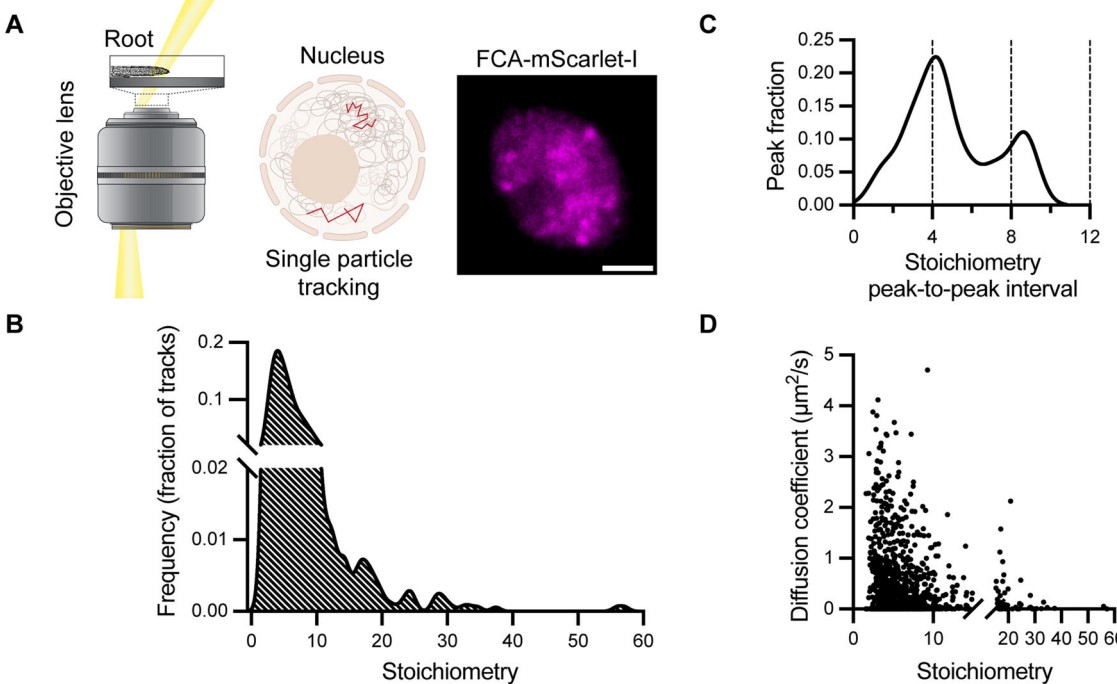

**Figure 1. Super-resolution microscopy and single-particle tracking to determine the physical characteristics of FCA particles.**

(A) A schematic of SlimVar microscopy used for the imaging Arabidopsis root cells and an example of a SlimVar image of a nucleus showing FCA-mScarlet-I (magenta). Scale bar: 2 µm. (B) Distribution of the stoichiometry (molecule number) of individual FCA particles. (C) Periodicity of FCA particle stoichiometry estimated from the most common spacing between neighbouring peaks in each stoichiometry distribution. (D) The diffusion coefficient and stoichiometry of each FCA particle. The single-particle tracking analysis is from three biologically independent experiments. Results from individual replicates are provided in Appendix Fig. S3. Source data are available online for this figure.

influence the in vivo biophysical properties (Bienz, 2020). Here, we therefore generated stable transgenic plants with fusions expressed under their own regulatory regions and chose lines that expressed the fusion at endogenous levels. Determining the biophysical properties of condensates in vivo in plant systems is challenging so we exploited SlimVar microscopy, a single-molecule imaging technique (Plank et al, 2009; Payne-Dwyer et al, 2024). This enables simultaneous tracking of individual particles within a wide range of biological condensates providing quantitative measurements of molecule number and molecular motion behaviour (Shen et al, 2023). This method has been used to study biomolecular condensates in live cells including bacteria, yeast or mammalian cells (Kent et al, 2020; Izeddin et al, 2014; Ladouceur et al, 2020; Biswas et al, 2022) and we adapted it for living plant tissues to specifically study nuclear proteins (Bayle et al, 2021).

Our findings indicate that FCA-mScarlet-I particles multimerize and then assemble in an FLL2-dependent manner into the liquid-like condensates observed using confocal microscopy. FLL2 and FCA predominantly co-localize in the larger condensates. The involvement of RNA in promoting FCA condensation was shown through the identification of a missense mutation in the FCA second RRM domain. This mutation reduced RNA-binding activity in vitro, significantly attenuated FCA particle size and condensate formation, lowered RNA 3' processing efficiency and prevented *FLC* repression but did not influence formation of the core FCA particle. Our work points to a modular structure for RNA 3' processing FCA condensates, with a low-order oligomer core that multimerizes into higher-order particles via associated RNA and

into larger functionally important condensates through FLL2 interaction.

# Results

## Single-particle tracking reveals physical characteristics of FCA particles

For the SlimVar microscopy, we chose to generate an FCA-mScarlet-I translational fusion as a fluorescent reporter. mScarlet-I is a bright red fluorescent protein known for its characteristics as a rapidly maturing monomer (Bindels et al, 2016). A mScarlet-I coding sequence was inserted into an FCA genomic clone and transgenic plant lines were generated, where the fusion was expressed at similar level as the endogenous FCA gene (Fig. EV1A,B). The FCA-mScarlet-I fusion formed nuclear foci (Figs. 1A and 3A) and rescued the *fca-9* mutant phenotype (Fig. EV1B), just like the FCA-eGFP fusion (Fang et al, 2019). We exploited SlimVar microscopy with a narrow, oblique illumination angle of 60 ± 5° to enhance imaging contrast and dynamically track very small FCA particles in nuclei in whole root tissue (Fig. 1A and Movie EV1). The single-molecule sensitivity of SlimVar in vivo, coupled with stepwise photobleaching of fluorescent protein tags (Leake et al, 2006; Reyes-Lamothe et al, 2010; Payne-Dwyer et al, 2024), allowed us to characterize single-molecule brightness of the FCA-mScarlet-I. This enabled quantification of the number of

FCA-mScarlet-I molecules (stoichiometry) within the particles in intact tissue (Fig. EV2 and Movie EV2; Jin et al, 2021).

Single-particle tracking was undertaken on homozygous transgenic plants carrying the FCA-mScarlet-I transgene expressed at endogenous levels in three biological replicate experiments. Analysis of the FCA-mScarlet-I stoichiometry distribution identified a distinct peak representing a low-order oligomer of FCA-mScarlet-I (Fig. 1B) with other peaks in multiples of approximately 4, from approximately 16 to 56 molecules (Fig. 1C). This finding aligns with a previous study that described the prion-like domain of FCA, which when fused to GFP in yeast formed small oligomers (Chakrabortee et al, 2016). To further investigate the oligomerization properties of FCA molecules we analysed in vitro produced FCA protein. Purified recombinant FCA proteins containing either the NH2-terminal half containing RRM domains (MBP-RRM, Fang et al, 2019) or the C-terminal FCA half containing prion-like domains (MBP-PrLD)—shown schematically in Fig. EV3—were analysed using size exclusion chromatography with multi-angle static light scattering (SEC-MALS). The FCA RRM fragment forms low-order oligomers containing 4 or possibly 5 FCA proteins, consistent with the particle tracking assay in vivo. The prion-like C-terminal domain formed very large aggregates with more than 100 molecules (Fig. EV3), a characteristic frequently associated with in vitro expression of PrLDs at relatively high concentrations.

SlimVar also enabled us to analyse the in vivo diffusion coefficient of individual particles with different stoichiometry. Generally, particles with low stoichiometry displayed variable mobility, while those with high stoichiometry showed low mobility (Fig. 1D). In particular, the diffusion coefficient of particles with stoichiometry less than 10 or more than 10 was $0.54 \pm 0.75$ μm²/s (mean ± s.d.) and $0.27 \pm 0.4$ μm²/s, respectively.

## FCA and FLL2 show co-localization in a larger size class of condensates

Our previous study using transient assays suggested that FCA condensates serve as the sites for the interaction between FCA, FLL2 and 3'-end processing factors (Fang et al, 2019). However, the condensates formed by FCA and FLL2 exhibited different patterns in low-resolution images of Arabidopsis roots (Fang et al, 2019). To investigate the potential co-localization of FCA and FLL2 within the same nuclear condensates in vivo, we crossed plants carrying an FCA-mTurquoise transgene (Xu et al, 2021) with plants carrying the FLL2-eYFP transgene (Fang et al, 2019) and used F3 plants expressing both reporters for imaging. 3D imaging and Airyscan confocal microscopy were used, acknowledging that the Z-axis resolution of the confocal microscope and the limitations of the imaging algorithm may underestimate the number and size of the condensates. We found that FCA-mTurquoise and FLL2-eYFP foci generally reside in distinct condensates within the cell nucleus (Fig. 2A; Appendix Fig. S1) but co-localize within a few larger condensates (Fig. 2B,D). We conducted a comprehensive analysis to determine the size distributions and the frequency of FCA, FLL2 and FCA/FLL2-co-localized condensates (Fig. 2C; Appendix Fig. S2). The sof78 missense mutation in FLL2 is predicted to influence a salt bridge connecting two coiled coils, and this attenuates FCA condensation and R-loop resolution (Fig. EV4; Fang et al, 2019). We used single-particle tracking to analyse whether sof78 affected the assembly of the low-order oligomers of FCA. The analysis reveals that the periodicity and stoichiometry distribution of FCA

particles in sof78 shows no obvious difference compared to the control (Fig. EV4B–D). Overall, the data suggest that the association of FLL2 promotes formation of the larger functional FCA condensates.

To address if these prominent FCA/FLL2 condensates directly associate with the FLC locus, we crossed the FCA-mScarlet-I transgene into a line carrying a previously established FLC-lacO/ eYFP-LacI system enabling spatial localization of the FLC locus in the nucleus. This line also carries an active FRIGIDA allele so FLC is in an active transcriptional state (Rosa et al, 2013). FCA condensates did not frequently co-localize with the FLC-lacO/ eYFP-LacI marked locus (Fig. EV5). This result suggests that FCA condensates may only transiently associate with the FLC locus in this genotype and environmental conditions. However, it will be interesting to repeat this analysis in a genotype where FLC is transcriptionally repressed by FCA, but where the Polycomb silencing has not yet been established (Menon et al, 2024).

## FCA condensation is dependent on a functional RRM domain

To further investigate the functional significance of the molecular composition in FCA condensates we continued a genetic screen for factors essential for FCA function (Liu et al, 2007). We used transgenic plants with overexpressed 35S::FCAγ and conducted a forward genetic analysis to identify mutations that suppressed FCA's ability to repress FLC expression. This identified a missense mutation on the FCA overexpression transgene at amino acid 304 of the FCA protein, resulting in a leucine to phenylalanine change (Fig. EV1C). This alteration is located within the second RRM domain of FCA, a domain conserved across various species (Macknight et al, 1997; Fig. EV1D). That the mutant came out of an FLC mis-expression screen and exhibited late flowering shows the RNA-binding activity is important for FCA function (Fig. EV1E). The impact of this missense mutation on the reduction of RNA-binding activity was established through in vitro RNA-binding assays (Fig. EV1F).

To examine whether this RRM mutation affects FCA condensation, we generated transgene constructs expressing an FCA wild-type genomic DNA fragment, with (FCArrm) and without (FCAwt) the mutation fused with mScarlet-I and introduced them into fca-9 mutant (Fig. EV1A). After selecting for single insertion transgenic plants and homozygotes, we analysed the FCAwt condensates within the nuclei of young roots using an Airyscan confocal microscope. The FCArrm transgene produced smaller, more diffuse condensates at lower numbers, compared to wild type (Fig. 3A,B).

We also conducted single-particle tracking with the FCArrm lines and observed the major oligomer peak was slightly higher than that for FCAwt (Fig. 3C; Appendix Fig. S3). FCArrm particles with more than 16 molecules were scarce, however, the integrated nuclear intensity of fluorescent signal in FCArrm and FCAwt showed no obvious change (Appendix Fig. S4). This result suggests that the reduced RNA-binding activity of FCA proteins decreases their likelihood of forming the larger particles in the range of 16 to 56 molecules. We also compared the diffusion coefficients of individual particles with their stoichiometry (Fig. 3D). The mean stoichiometry of FCArrm was lower than that of FCAwt, while the overall the diffusion coefficients were slightly higher (Fig. 3E; Appendix Fig. S5), however, the periodic unit was not significantly changed (Fig. 3F). Thus, the RNA-binding activity of FCA proteins

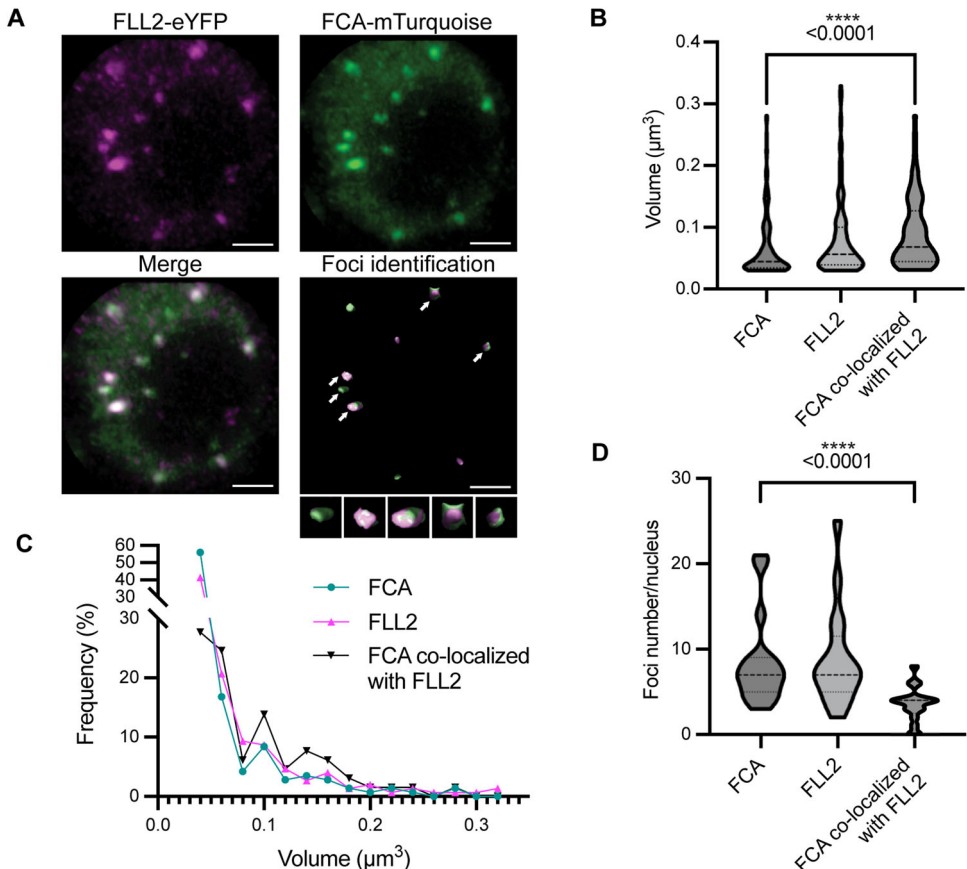

**Figure 2. The co-localization of FCA and FLL2 is associated with larger-sized condensates in Arabidopsis nuclei.**

(A) Selected 3D images of FCA and FLL2 colocalization. FCA-mTurquoise signal (green) and FLL2-eYFP signal (magenta) were segmented using the blob finder algorithm of Arivis Vision4D for foci identification, as described in the Methods section. White arrows indicate the co-localized condensates. The enlarged images of co-localized foci are shown as insets at the bottom. Scale bars: 2 μm. (B) Quantification of size of FCA (total FCA-mTurquoise identified foci), FLL2 (total FLL2-eYFP identified foci) and FCA identified foci that are co-localized with FLL2. The lines in the violin plots indicate the median and quartiles. FCA vs. FCA co-localized with FLL2, P-value = 2.4e−05. (C) Size and frequency distribution of FCA, FLL2 foci and FCA foci that are co-localized with FLL2. (D) Number of FCA, FLL2 foci and FCA foci that are co-localized with FLL2 in each nucleus. FCA vs. FCA co-localized with FLL2, P-value = 1.1e−06. The lines in the violin plots indicate the median and quartiles. The quantitative analysis was based on 18 nuclei from 5 independent samples. Statistical significance was determined using non-parametric Brunner-Munzel test. ****P-value < 0.0001. Source data are available online for this figure.

affects multimerization of the FCA particles, and formation of the larger condensates, but does not affect core oligomer assembly.

### The functional RNA-binding domain is required for *COOLAIR* R-loop resolution and FCA auto-regulation

We then used the FCAwt and FCArrm lines to further investigate the functional importance of the RNA-binding domain. Compared to the FCAwt transgenic plants, FCArrm plants displayed higher levels of *FLC* expression and therefore flowered late (Fig. 4A,B and EV1B). FCA associates with *COOLAIR*, recruiting 3' processing factors to promote proximal *COOLAIR* polyadenylation, thus resolving a *COOLAIR*-formed R-loop (Liu et al, 2010; Xu et al, 2021). The ratio of proximal-to-distal isoforms of *COOLAIR* transcripts was reduced in the FCArrm lines compared to FCAwt (Fig. 4C). Consistent with this result, S9.6-DNA/RNA immunoprecipitation followed by cDNA conversion (DRIPc)-qPCR analysis showed higher R-loop level in FCArrm relative to FCAwt, indicating that FCArrm has reduced ability to resolve the *COOLAIR*-induced R-loop (Fig. 4D).

FCA autoregulates itself promoting polyadenylation at the proximal sites in intron 3 (Quesada et al, 2003), so we conducted Quant-seq analysis (Moll et al, 2014) to capture polyadenylated *FCA* transcripts in Col-0, FCAwt, FCArrm and *fca-9*. In FCArrm and *fca-9*, there were significantly fewer reads corresponding to proximally polyadenylated *FCA* transcripts compared to FCAwt and Col-0. Conversely, reads at the distally polyadenylated site of *FCA* increased (Fig. 5A,B), all consistent with reduced FCA function. RNA-binding activity is thus important for FCA particle multimerization and formation of the condensates, and for the well-defined FCA functional roles; promotion of proximal *FCA* and *COOLAIR* termination and R-loop resolution.

## Discussion

Our study provides insight on the functional properties of the dynamic liquid-like RNA 3' processing condensates containing the Arabidopsis nuclear RNA-binding protein FCA. We conducted

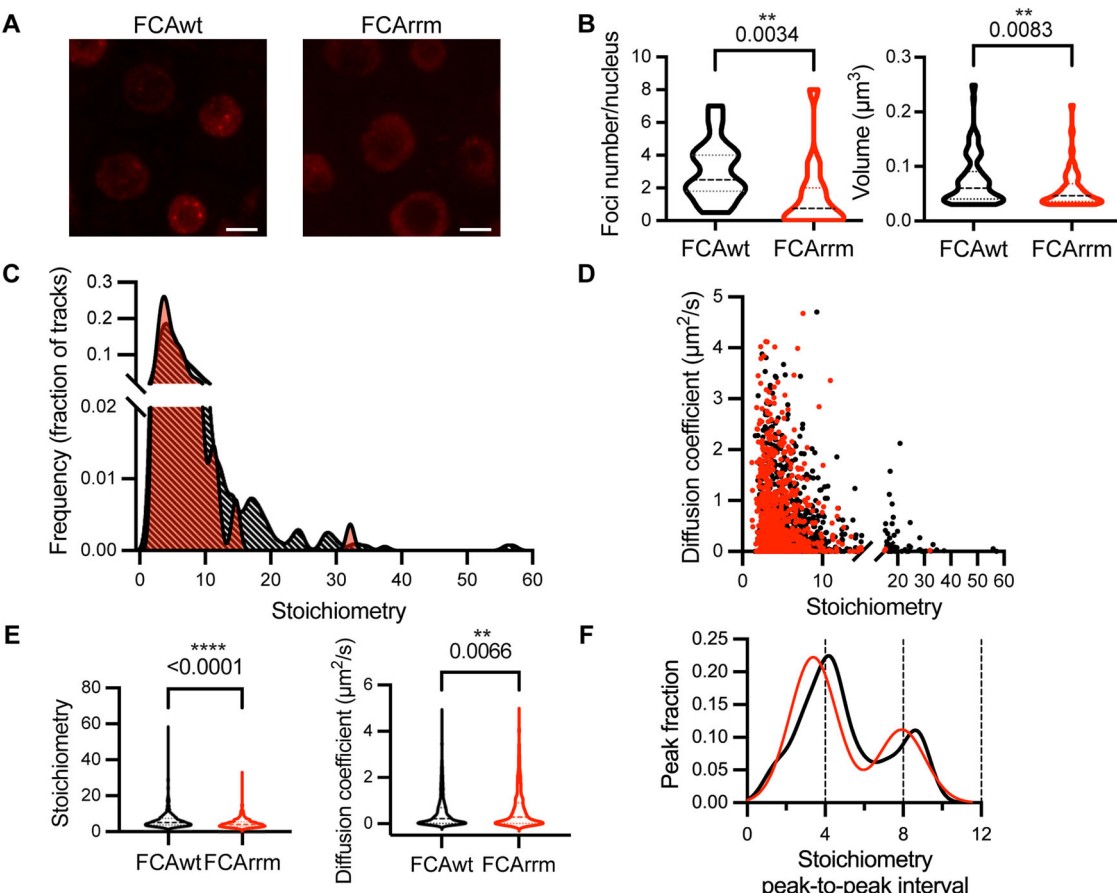

**Figure 3. The RNA-binding activity of FCA influences FCA condensate size and number.**

(A) Selected Airyscan confocal images of wild-type FCA fused to mScarlet-I (FCAwt) and FCA-mScarlet-I with the missense (L304F) mutation (FCArrm). Scale bars: 5 μm. (B) Quantification of size and number of FCAwt and FCArrm condensates using Vision4D software. The lines in the violin plots indicate the median and quartiles. The quantitative analysis is based on 43 FCAwt nuclei and 65 FCArrm nuclei from 3 independent experiments. (C) Distributions of stoichiometry of individual FCAwt (black) and FCArrm (red) particles. (D) Diffusion coefficient and stoichiometry of each FCAwt (black) and FCArrm (red) particle. (E) Statistical analysis of stoichiometry and diffusion coefficient of FCAwt ($n = 1008$) and FCArrm ($n = 913$) particles. Stoichiometry of FCAwt vs. FCArrm, $P$-value $= 2e-29$. (F) Periodicity of FCAwt (black) and FCArrm (red) particle stoichiometry. The single-particle tracking analysis is from 3 biologically independent experiments. The FCAwt data corresponds to that shown in Fig. 1. Statistical significance was determined using non-parametric Brunner-Munzel test. **$P$-value $< 0.01$, ****$P$-value $< 0.0001$. Results of single-particle tracking from individual replicates are provided in Appendix Figs. S3 and S5. Source data are available online for this figure.

single-particle tracking to characterize molecular stoichiometry and diffusivity, and the influence of protein and RNA partners that function in *FLC* repression. Our work points to a modular structure for FCA condensates centred around a low-order core oligomer, which multimerize to oligomers and macromolecular condensates via associated RNA and FLL2 interaction. The genetic data support a requirement for these condensates in FCA function for RNA 3' processing (Figs. 3A, 4 and EV4A; Fang et al, 2019) with RNA-protein and protein-protein interactions playing different roles in the oligomerization and condensation processes.

The FCA protein has two RRM domains (Macknight et al, 1997) and an NH2 terminus that initiates translation with a CTG (non-methionine) start codon (Simpson et al, 2010). In the C-terminal half of FCA there is an extended prion-like domain (Chakrabortee et al, 2016) interrupted by a WW protein interaction domain (Simpson et al, 2003; Henderson et al, 2005). SEC-MALS analysis showed the N-terminal RRM fragment multimerizes in vitro into ~4 mers. Notably, this is reminiscent of ELAV, a well-characterized Drosophila RNA-binding

protein, which forms a tetramer in the absence of substrate RNA (Soller and White, 2005). Thus, the N-terminal RRM region potentially determines the basal tetrameric state of the FCA protein, with aggregation of the prion-like domain prevented in vivo. Further work will be required to fully determine how all these domains influence FCA condensate dynamics. For FCA condensates, the role of RNA may involve functionality of the hydrophobic core in the RRM domain. The missense mutation (L304F) could strengthen the hydrophobic core in the FCA second RRM domain, an equivalent hydrophobic region to that recently found to influence the protein stability of TDP-43, an amyotrophic lateral sclerosis (ALS)-linked RNA binding protein (Mackness et al, 2024). RNA methylation may also be a contributing factor to the condensation dynamics as m6A has been shown to function in the regulatory mechanisms of *FLC* and *COOLAIR* (Xu et al, 2021; Wang et al, 2022; Sun et al, 2022; Amara et al, 2023).

*COOLAIR* transcription is enhanced upon cold exposure, and the transcripts form dense 'clouds' at each *FLC* locus, as judged by single-molecule RNA fluorescence in situ hybridization (Rosa et al,

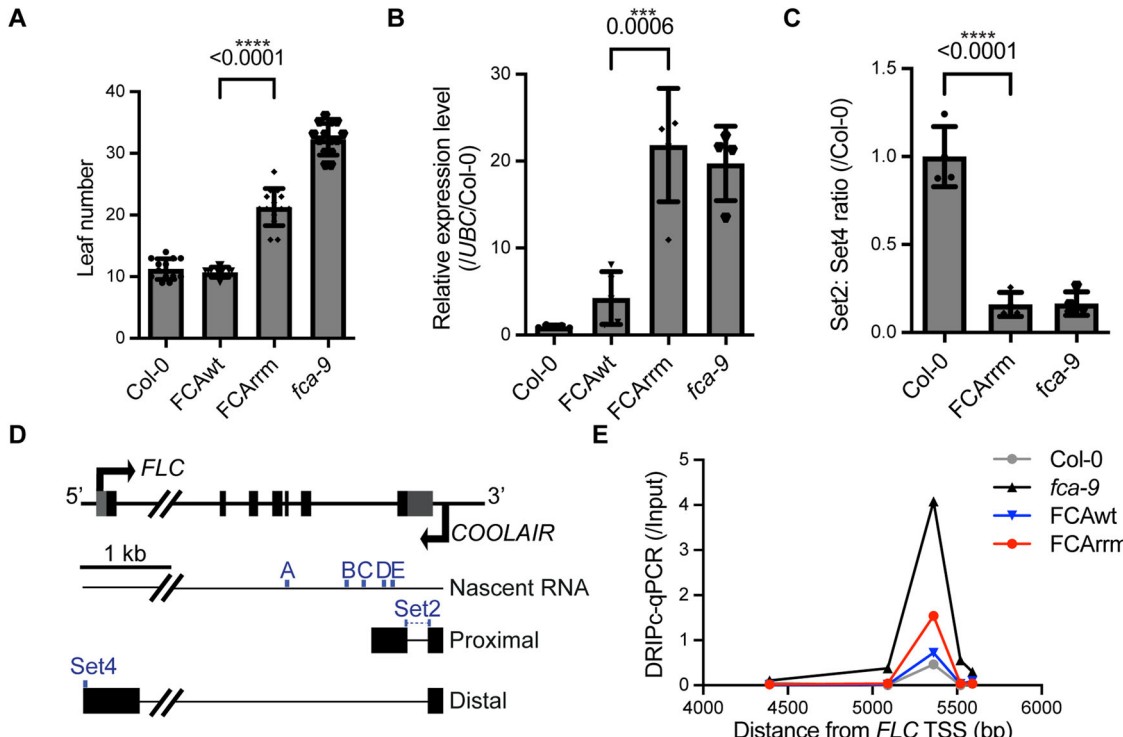

**Figure 4. A missense mutation in the RRM of FCA only partially complements the *fca-9* mutant phenotype.**

(A) Flowering time of selected transgenic lines (#8 for FCAwt and #1 for FCArrm in Fig. EV1) grown in a long-day photoperiod. Data are presented as mean ± s.d. (*n* = 14 seedlings, except for Col-0 which *n* = 13). FCAwt vs. FCArrm, *P*-value = 1.1e−12. (B) Relative values of *FLC* in the indicated lines. Values were normalized to the *UBC* gene and to Col-0. Data are presented as mean ± s.d. (*n* = 5 biological replicates, except for *fca-9* where *n* = 4). (C) The ratio of proximal: distal isoforms of *COOLAIR* transcripts relative to control (Col-0) in the indicated plants. Data are presented as mean ± s.d. (*n* = 4 biological replicates). Col-0 vs. FCArrm, *P*-value = 9.7e−05. (D) Schematic of *FLC* and *COOLAIR* transcripts at the *FLC* locus. Blue dots indicate the locations of primer sets for qPCR. Untranslated regions are indicated by grey boxes, and exons by black boxes. kb, kilobase. (E) DRIPc–qPCR analysing *COOLAIR* R-loop in indicated genotypes. Data are presented as mean from four technical repeats. Three biological replicates show similar trends (Appendix Fig. S7). TSS, transcription start site. The primer sets used for DRIPc-qPCR (A–E) corresponds to that shown in (D). Statistical significance was calculated using the two-tailed t-test, with *P*-values indicated. ***\*P*-value < 0.001, \*\*\*\**P*-value < 0.0001. Source data are available online for this figure.

2016). Whether the FCA condensates associate with these *COOLAIR* clouds will require RNA live imaging analysis. It is interesting to speculate that *COOLAIR* molecules might undergo thermo-responsive phase transitions as has been found recently for other RNAs (Wadsworth et al, 2023), with protein partners tuning this phase behaviour (Ruff et al, 2021). The *FLC* biological system has a lot of potential to uncover fundamental principles of condensate functionality in vivo, especially if we can successfully image the condensates at the locus. In mammalian cells, Mediator conden-sates only sporadically co-localize with *Sox2*, and this variability is attributed to a dynamic kissing model, where a Mediator condensate interacts with the gene only at specific timepoints (Cho et al, 2018; Du et al, 2024). This may be the case for *FLC*, which can exist in either a transcriptionally active (with functional FRIGIDA) or inactive (without functional FRIGIDA) state; with switching between these states connected to co-transcriptional RNA processing (Fang et al, 2020; Mateo-Bonmati et al, 2024; Menon et al, 2024). Exploring whether the transient association of FCA condensates with *FLC* locus reflects the transcriptional status of *FLC* will be the next important question.

In this study, we used SlimVar microscopy, which integrates Slimfield's technical capabilities with enhanced imaging contrast from variable-angle laser excitation (Plank et al, 2009; Payne-Dwyer et al, 2024). In contrast to a previous approach using a combination of Airyscan and Slimfield to infer the stoichiometry of FCA assemblies indirectly (Payne-Dwyer et al, 2022), SlimVar allowed for direct and precise single-particle tracking and stoichiometric quantification of molecular assemblies in plant root tissue. The oblique excitation beam reduces out-of-focus background fluorescence, similar to HILO and lightsheet microscopy, and the use of an oil immersion objective lens improves fluorescence signal despite some optical aberration (Toku-naga et al, 2008; Stelzer et al, 2021). SlimVar calibration techniques such as adjusting the objective lens correction collar helped minimize these aberrations, enabling effective postprocessing "sifting" to collect robust signals (Payne-Dwyer et al, 2024). The limitations of SlimVar microscopy when observing FCA particles included the typically small field of view, which usually captures just one nucleus per acquisition. Additionally, the methodology suffered from fewer or shortened tracks due to bleaching, diffusion of assemblies, and residual aberration and scattering, necessitating large sample numbers. Furthermore, at higher assembly concentrations, careful image analysis was required to avoid overlapping assemblies leading to incorrect stoichiometry estimates. Despite these limitations, SlimVar provides the ability to functionally study particles involved in biomolecular condensation in Arabidopsis.

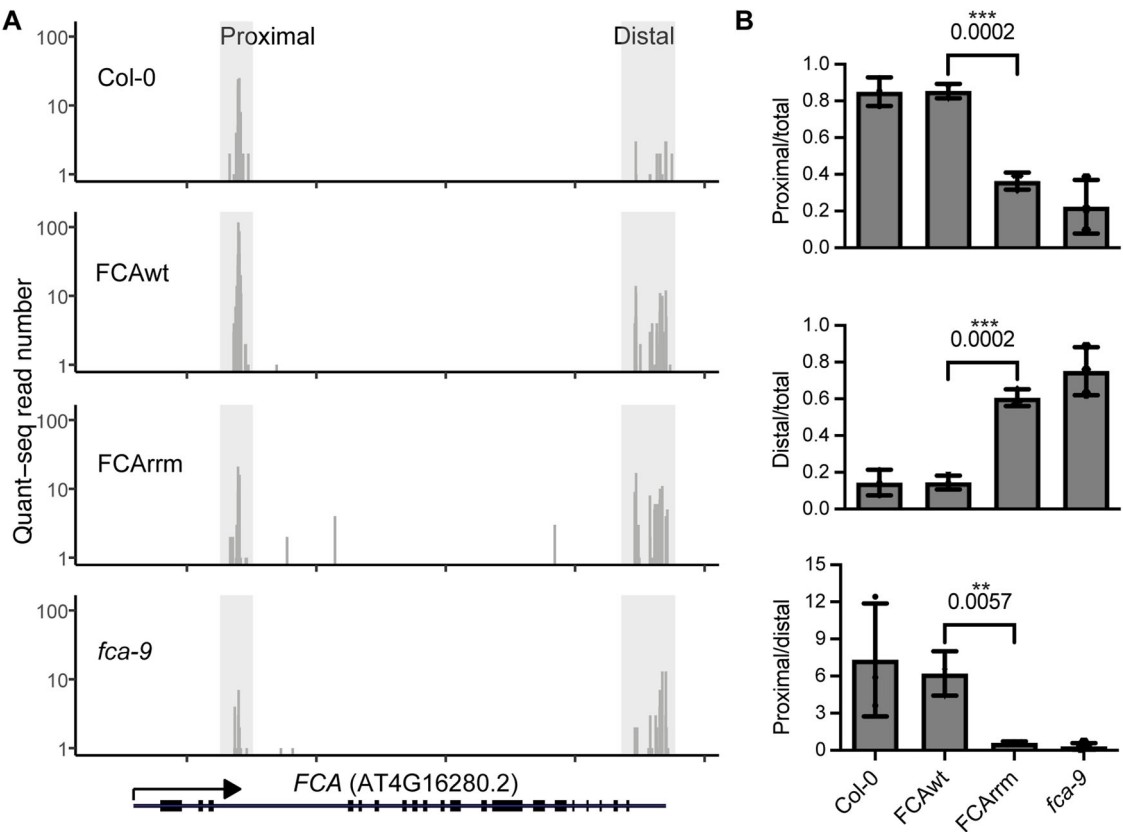

**Figure 5.  The RRM mutation perturbs the autoregulatory function of FCA in promoting proximal polyadenylation.**

(A) Quant-seq experiments assaying polyA sites at the *FCA* locus in different genotypes. The vertical pale grey bars highlight previously described proximal and distal polyA sites. (B) Quantification of reads corresponding to proximal and distal polyadenylated *FCA* transcripts. Data are presented as mean ± s.d. ($n = 3$ biological replicates). Statistical significance was calculated using the two-tailed t-test, with *P*-values indicated. **P*-value < 0.01, ****P*-value < 0.001.

This opens up the possibility of combining a genetically tractable system with dissection of condensate function and their dynamic responses to various environmental exposures.

# Methods

### Reagents and tools table

| Reagent/Resource | Reference or Source | Identifier or Catalog Number |
|---|---|---|
| **Experimental models** | | |
| *Arabidopsis thaliana* Col-0 | Standard accession | |
| *A. thaliana fca-9* | Fang et al, 2019 | |
| *A. thaliana pFLL2::FLL2-eYFP* | Fang et al, 2019 | |
| *A. thaliana pFCA::FCA-eGFP* | Fang et al, 2019 | |
| *A. thaliana pFCA::FCA-eGFP sof78* | Fang et al, 2019 | |
| *A. thaliana pFCA::FCA-mTurquoise* | Xu et al, 2021 | |

| Reagent/Resource | Reference or Source | Identifier or Catalog Number |
|---|---|---|
| *A. thaliana FLC-lacO/eYFP-LacI* | Rosa et al, 2013 | |
| *A. thaliana pFCA::FCA-mScarlet-I* | This study | |
| *A. thaliana pFCA::FCA-L304F-mScarlet-I* | This study | |
| **Recombinant DNA** | | |
| Plasmid: pSLJ755I5 | Jones et al, 1992 | |
| Plasmid: pENTR-*pFCA::FCA-mScarlet-I* | This study | |
| Plasmid: pENTR-*pFCA::FCA-L304F-mScarlet-I* | This study | |
| Plasmid: pSLJ-*pFCA::FCA-mScarlet-I* | This study | |
| Plasmid: pSLJ-*pFCA::FCA-L304F-mScarlet-I* | This study | |
| **Antibodies** | | |
| S9.6 antibody | Kerafast | ENH001 |

| Reagent/Resource | Reference or Source | Identifier or Catalog Number |
|---|---|---|
| **Oligonucleotides and other sequence-based reagents** | | |
| Primers | This study | Table EV1 |
| **Chemicals, Enzymes and other reagents** | | |
| Phusion High-Fidelity DNA Polymerases (2 U/μL) | Thermo Fisher Scientific | F530S |
| dNTP Set (100 mM) | Thermo Fisher Scientific | 10297018 |
| GoTaq G2 Flexi DNA Polymerase | Promega | M7801 |
| Gateway LR Clonase II Enzyme mix | Thermo Fisher Scientific | 11791020 |
| cOmplete, EDTA-free Protease Inhibitor Cocktail | Roche | 11873580001 |
| In-Fusion Snap Assembly Master Mix | Takara | 638948 |
| LightCycler 480 SYBR Green I Master | Roche | 04887352001 |
| Phenol solution saturated with 0.1 M Citrate | Merk Life Science | P4682 |
| RNaseOUT Recombinant Ribonuclease Inhibitor | Thermo Fisher Scientific | 10777019 |
| SuperScript IV Reverse Transcriptase | Thermo Fisher Scientific | 18090050 |
| T4 DNA Ligase | New England Biolabs | M0202S |
| Agarose molecular biology grade | Melford | A20080 |
| Murashige & Skoog Medium including vitamins | Duchefa | M0222 |
| 0.2 μm TetraSpeck microspheres | Thermo Fisher Scientific | T7280 |
| proteinase K | Thermo Fisher Scientific | AM2546 |
| DdeI | New England Biolabs | R0111L |
| NdeI | New England Biolabs | R0175 |
| XbaI | New England Biolabs | R0145L |
| RNAse III | Thermo Fisher Scientific | AM2290 |
| Dynabeads Protein G for Immunoprecipitation | Thermo Fisher Scientific | 10004D |
| RNAsecure RNase Inactivation Reagent | Thermo Fisher Scientific | AM7005 |
| **Software** | | |
| Geneious Prime 2024.0.4 | https://www.geneious.com | |
| Zen | Zeiss | |
| Arivis Vision4D ver. 4.1.0. | Zeiss | |
| GraphPad Prism 10 | GraphPad Software www.graphpad.com | |
| ADEMScode | Plank et al, 2009 | |
| MATLAB | MathWorks https://ch.mathworks.com/products/matlab.html | |
| Wyatt ASTRA 7.3.2 | Wyatt Technology https://www.wyatt.com/products/software/astra.html | |

| Reagent/Resource | Reference or Source | Identifier or Catalog Number |
|---|---|---|
| LightCycler480 II | Roche | |
| cutadapt version 1.18 | Martin, 2011 | |
| splice-aware aligner STAR version 2.6.1a | Dobin et al, 2013 | |
| **Other** | | |
| TURBO DNA-*free* Kit | Thermo Fisher Scientific | AM1907 |
| Qubit dsDNA Quantification Assay Kits | Thermo Fisher Scientific | Q32850 and Q32851 |
| Qiagen RNeasy miniprep kit | QIAGEN | 74106 |
| MAXIscript T7 Transcription Kit | Thermo Fisher Scientific | AM1312 |
| Pierce RNA 3′ End Biotinylation Kit | Thermo Fisher Scientific | 20160 |
| Chemiluminescent Nucleic Acid Detection Module Kit | Thermo Fisher Scientific | 89880 |

Instructions: Please complete the relevant fields below, adding rows as needed. The following page provides an example of a completed table and additional instruction for entering your data in the table.

## Plant materials and growth conditions

The wild type Col-0, *fca-9* mutant allele, *pFCA::FCA-mTurquoise* and *pFLL2::FLL2-eYFP* transgenic plants were described previously (Fang et al, 2019; Xu et al, 2021). To generate the *pFCA::FCA-mScarlet-I* transgenic plants, the FCA genomic DNA fragment was amplified and inserted into the pENTR vector. The mScarlet-I coding sequence was inserted before the stop codon via In-Fusion seamless cloning. The FCA genomic DNA fragment fused mScarlet-I was transferred to pSLJ755I5 vector (Jones et al, 1992) via gateway cloning. The RRM mutation construct is modified from wild-type FCA genomic DNA constructs via In-Fusion seamless cloning. Primers were listed in Table EV1. These constructs were transformed into the *fca-9* mutant. Arabidopsis seeds were surface sterilized and sown on standard Murashige and Skoog (MS) medium plate without glucose and stratified at 4 °C for 3 days before transferred to long-day conditions (16-h light at 20 °C, 8-h darkness at 16 °C) for 4–7 days.

## Confocal microscopy of FCA-mScarlet-I

Confocal images were acquired on a Zeiss LSM 880 upright microscope equipped with an Airyscan detector. Samples were imaged through a C-Apochromat 63x/NA 1.2 water-immersion objective by exciting them at 561 nm and using the Airyscan detector in super resolution mode. The fluorescence emission arrived at the detector after passing through a filter that allowed wavelengths between 495–550 nm and above 570 nm to pass through. Typically, the voxel size was $0.06 \times 0.06 \times 0.26\ \mu m$ (xyz) and the pixel dwell time was 1.78 μs. Z-stacks typically comprised 28 slices and spanned a 7.13 μm range for more than 70% of the nucleus. In Fig. 3A, the FCA-mScarlet-I signal was enhanced by shifting its dynamic range to 0–160, and the images were processed using maximum intensity projection. After Airyscan processing, the

image analysis was performed in Arivis Vision4D ver. 4.1.0 (Zeiss). The blob finder algorithm was applied to the mScarlet-I channel using a diameter value of 0.35 μm, a probability threshold of 45%, and a split sensitivity of 50%. Blob finder is a segmentation algorithm that excels at finding round or spherical objects in 2D or 3D, respectively. It uses a Gaussian scale to identify seeds for the objects and subsequently applies a watershed algorithm to determine the boundaries of the objects. To exclude non-specific identified condensates, a minimum size of 0.03 μm³ and a sphericity (Mesh) of 0.6 were set during the analysis, and non-specific foci outside the nuclear region of interest were removed. Violin plots reflecting the data distribution were produced with GraphPad Prism 10 software.

## Co-localization analysis of FCA and FLL2

Confocal images were acquired either on the Zeiss LSM 880 microscope mentioned above or a Zeiss LSM 980 confocal microscope. When using the LSM880, line by line sequential acquisition was employed. Samples were imaged through a C-Apochromat 63x/NA 1.2 water-immersion objective and were excited at 458 nm (mTurquoise) and 514 nm (eYFP). The fluorescence emission of the samples passed through a filter that only allowed wavelengths in the 465–505 nm range and above 525 nm to reach the Airyscan detector, which was operating in super-resolution mode. The voxel size was 0.05 × 0.05 × 0.21 μm (xyz) and the pixel dwell time was 0.72 μs. Z-stacks comprised 25 slices spanning a 4.96 μm range. 0.2 μm TetraSpeck microspheres (Thermo Fisher Scientific) were imaged using these same settings and the resulting images were used to correct the sample images for chromatic aberration. The correction was performed in Zen (Zeiss) using the channel alignment method. In Fig. 2A, FCA-mTurquoise and FLL2-eYFP signal was enhanced by shifting the dynamic range of the signal to 0–1650 and 0–750.

After Airyscan processing, channel alignment, and nuclei segmentation, the image analysis was performed in Arivis Vision4D ver. 4.1.0 (Zeiss). Firstly, the blob finder algorithm was applied to the mTurquoise channel using a diameter value of 0.3 μm, a probability threshold of 37%, and a split sensitivity of 50%. Then, the blob finder algorithm was applied to the eYFP channel using a diameter value of 0.35 μm, a probability threshold of 37%, and a split sensitivity of 50%. Afterwards, the intersection between the output of the two blob finder operations was calculated. Finally, metrics such as volume, mean intensity, and total intensity were computed for the objects generated by each of the blob finder operations, as well as for their intersection. In general, a minimum size of 0.03 μm³ and a sphericity (Mesh) of 0.6 were set during the analysis, and non-specific foci outside the nuclear region of interest were removed. Violin plots and x–y plots reflecting the data distribution were produced with GraphPad Prism 10 software.

## SlimVar microscopy

For all imaging modalities, seedlings' root tips were oriented horizontally on slides containing MS media with 1% agarose, sealed with MS media under a No. 1.5 coverslip. The customized SlimVar microscope was previously described (Payne-Dwyer et al, 2024). Continuous wave lasers (Coherent OBIS) delivered a de-expanded Gaussian beam mode (TEM$_{00}$) to the back aperture of an NA 1.49 Apo TIRF 100× oil immersion objective lens (Nikon) with a lateral

displacement such that the beam illuminated a sample area no greater than $9 \times 16$ μm at $60 \pm 5°$ incidence. Single-molecule sensitivity was achieved using a fast sCMOS camera (Photometrics Prime95B) which triggered a continuous wave laser for 10 ms exposure per frame at 80 Hz, providing optimal contrast for detection (Appendix Fig. S6). The total magnification was approximately 200×, resulting in an oversampled image pixel edge size of 53 nm. Excitation and emission filters were tailored to best contrast for mScarlet-I (594/25 nm). Fluorescence acquisition settings in the red and complementary green channels were optimized to prevent initial saturation and to ensure the detection of individual molecular brightness tracks for mScarlet-I. To calibrate SlimVar, the beam displacement lens and objective lens correction collar were adjusted in situ to minimise optical aberrations at a representative 20 μm depth. These settings were then locked to prevent systematic variation in the characteristic single-molecule brightness. Individual nuclei in the outermost 3 cell layers were identified in brightfield mode to determine the best focus without bleaching, and subsequently imaged in the SlimVar fluorescence mode until completely photobleached. Videos were subject to a temporal bleach correction (Fiji, Schindelin et al, 2012). Gaussian smoothing of 0.5 pixels (Fiji) were used on images, min/max adjusted for contrast, and the black point set at the background level outside the nucleus.

## Single-particle tracking and intensity analysis

Fluorescent particles were identified from SlimVar image sequences and linked into tracks using ADEMScode software (Plank et al, 2009) using a strict 'sifting' approach. This sets signal-to-noise and distance thresholds to discard foci with low contrast, whose integrated intensity is influenced by background or detection noise, or exhibit nonphysical mobility. Each of these particles was localised to a super-resolved, subpixel precision, typically 40 nm. The random overlap probability was estimated using a nearest neighbour distance approach as <5% for all observed tracks (Payne-Dwyer et al, 2024). Analysis was spatially restricted to data lying within a region of effectively uniform laser illumination (80% ± 9% s.d. peak intensity) by cropping images to an area of $240 \times 184$ pixels ($12.7 \times 9.8$ μm). Fluorescence image sequences were further cropped to specify individual nuclei, using manually segmented masks from brightfield images.

The 2D diffusion coefficient of each track ($D$) was computed using the increase of mean-square displacement across time interval according to a random walk model (Leake et al, 2008). Focus intensity was calculated by summing the pixel values within a 5-pixel distance and subtracting an averaged background level from the rest of a 17-pixel square (Leake et al, 2008). The track stoichiometry was found by extrapolating the particle's intensity in the first 5 image frames s to account for photobleaching (Syeda et al, 2019), then dividing this initial intensity by the characteristic single molecule brightness (Wollman et al, 2017). The characteristic single molecule brightness linked to each reporter was determined as 110 ± 15 (mean ± s.d.) for mScarlet-I based on the modal size of steps in track intensity during late-stage photobleaching. The track intensities over time were first denoised using an edge preserving filter (Chung and Kennedy, 1991). This characteristic single molecule brightness provides an internal calibration for track stoichiometry, periodicity, and integrated nuclear intensity which is refined to nuclear protein number.

To calculate periodicity, the stoichiometries of all tracks within each nucleus were represented as a kernel density distribution

(Leake et al, 2014), employing an empirical standard deviation of 0.6 molecules on the characteristic single molecule brightness. Peaks within this distribution were identified using the MATLAB *findpeaks* function, and the intervals between nearest neighbour peaks were computed (Payne-Dwyer et al, 2024). Raw estimations of the total number of molecules in each nucleus were determined using ImageJ macros to integrate the pixel values inside segmented nuclei as in previous work (Wollman and Leake, 2015; Payne-Dwyer et al, 2024). The nuclear protein number of each reporter dataset was refined to exclude autofluorescence by calculating the difference between mean integrated nuclear intensities of the labelled dataset and an unlabelled control, adjusted proportionally to the ratio of mean nuclear segment areas.

## In vitro protein expression and purification

MBP-RRM and MBP-PrLD proteins were expressed and purified from *Escherichia coli* (Rosetta) cells using Ni-NTA resin as described previously (Fang et al, 2019). Briefly, protein expression was induced by 0.4 mM isopropyl-β-D-1-thiogalactopyranoside (IPTG) at 18 °C overnight. Cells were collected by centrifugation and re-suspended in lysis buffer (40 mM Tris-HCl pH 7.4, 500 mM NaCl, 10% glycerol). The suspension was sonicated for 20 min (3 s ON, 3 s OFF, SCIENTZ) and centrifuged at $13,000 \times g$ for 30 min at 4 °C. The supernatant was incubated with Ni-NTA agarose for 20 min, washed and eluted with buffer E (40 mM Tris-HCl pH7.4, 500 mM NaCl, 500 mM Imidazole, Sangon). Proteins were further purified by gel filtration chromatography (Superdex-200; GE Healthcare) and stored in buffer S (40 mM Tris-HCl pH 7.4, 500 mM NaCl, 1 mM DTT) at −80 °C.

## Size-exclusion chromatography coupled with multi-angle laser light scattering (SEC-MALS)

Multi-angle light scattering experiments were performed in 40 mM Tris-HCl pH7.4 and 500 mM NaCl using a Superdex-200 10/300 GL size exclusion column (GE Healthcare). The concentration of MBP-RRM and MBP-PrLD used was 2 mg/mL. The chromatography system was connected to a Wyatt DAWN HELEOS laser photometer and a Wyatt Optilab T-rEX differential refractometer. Wyatt ASTRA 7.3.2 software was used for data analysis.

## Expression analysis

Total RNA was extracted from plant materials as described previously (Qüesta et al, 2016), treated with TURBO DNase (Thermo Fisher Scientific) to eliminate DNA contamination, and reverse-transcribed using SuperScript IV Reverse Transcriptase (Thermo Fisher Scientific) with gene-specific reverse primers. Quantitative PCR (qPCR) analysis was conducted on a Light-Cycler480 II (Roche), and the qPCR data were normalized to the reference genes *UBC*. To measure the proximal ratio of *COOLAIR*, the level of the proximal component was normalized to the distal *COOLAIR*. Primers were listed in Table EV1.

## DNA−RNA hybrid immunoprecipitation followed by cDNA conversion (DRIPc)

The DRIP protocol is as described previously (Xu et al, 2021) with modifications. In general, 3 g 10-day-old seedlings were harvested and grounded into a fine powder. The powder was suspended in 35 mL of Honda buffer (20 mM HEPES, 0.44 M sucrose, 1.25% Ficoll, 2.5% dextran T40, 10 mM MgCl$_2$, 0.5% Triton X-100, 5 mM DTT, 1x protease inhibitor cocktail (Roche), filtered through one layer of Miracloth, and centrifuged at $3500 \times g$ for 15 min. Nuclear pellets were resuspended in 1 mL Honda buffer and centrifuged at $8000 \times g$ for 1 min. Pellets were then resuspended in the lysis buffer (50 mM Tris-HCl pH 8.0, 10 mM EDTA, 1% SDS) supplied with 0.1 mg/mL proteinase K (AM2546, Thermo Fisher Scientific) and digested at 55 °C overnight with gentle rotation. The mixture was phenol/chloroform extracted, followed by nucleic acid precipitation with NaOAc and ethanol. The nucleic acid pellet was dissolved in water and quantified with Qubit DNA quantification kit (Thermo Fisher Scientific). In general, 1 μg of nucleic acid was digested by restriction enzymes including DdeI, NdeI and XbaI (NEB) overnight. The digested nucleic acid was phenol/chloroform extracted, followed by nucleic acid precipitation with NaOAc and ethanol. Then the nucleic acid was treated with RNAse III (Thermo Fisher Scientific) and then extracted by phenol/chloroform and ethanol precipitation. The treated nucleic acid was then diluted ten times with dilution buffer (16.7 mM Tris pH 7.5, 167 mM NaCl, 2.2 mM EDTA, 0.1% Triton X-100) and 10% was stored at −20 °C as input. In all, 5 μg of S9.6 antibody (1:100 dilution, ENH001, Kerafast) was added, then incubated overnight at 4 °C. The next day, 50 μl Dynabeads Protein G (Thermo Fisher Scientific) was added and incubated for another 2 h. The immuno-precipitants were washed five times with dilution buffer and twice with TE buffer, then were eluted in 100 μl elution buffer (10 mM Tris pH 7.5, 2 mM EDTA, 0.2% SDS, 100 ng/μl tRNA) and 5 μl proteinase K at 55 °C for 2 h, together with input samples. The nucleic acids were precipitated with NaOAc, isopropanol, and glycogen, dissolved in water. The DRIP nucleic acid samples were then treated with TURBO DNase (Thermo Fisher Scientific) to eliminate DNA contamination, and reverse-transcribed using SuperScript IV Reverse Transcriptase (Thermo Fisher Scientific) with gene-specific reverse primers. All the samples including inputs were subjected to qPCR analysis via LightCycler480 II (Roche). The data were normalized to 1% of input. Primers were listed in Table EV1.

## Quant-seq

The Quant-seq protocol is as described previously (Mateo-Bonmatí et al, 2024). The total RNAs prepared for Quant-seq underwent purification using the Qiagen RNeasy miniprep kit (QIAGEN, 74106). Lexogen GmbH (Austria) conducted the library preparation, sequencing, and data analysis. Sequencing-ready libraries were created from 100 ng of input RNA using a QuantSeq 3' mRNA-Seq Library Prep Kit REV for Illumina (015UG009V0271), following standard procedures. The RNA integrity and quality of indexed libraries were assessed on a Fragment Analyzer device (Agilent Technologies) using a DNF-471 RNA Kit and HS-DNA assay, respectively. Library quantification was carried out using a Qubit dsDNA HS assay (Thermo Fisher Scientific). A sequencing-ready pool of indexed libraries was then sequenced on an Illumina NextSeq 2000 with a 100-cycle cartridge, using the Custom Sequencing Primer (CSP). Verification of read quality utilized FastQC version v0.11.7, and read adaptor trimming employed cutadapt version 1.18 (Martin, 2011). Clean reads were mapped to the latest version of the Arabidopsis genome (TAIR10) using the splice-aware aligner STAR version 2.6.1a (Dobin et al, 2013).

## Electrophoretic mobility shift assay

FCA cDNA with L304F mutation was amplified from the mutant as described in Fig. EV1.

To generate RNA probes, dsDNA of the corresponding region was amplified through PCR (primers see Table EV1) with the T7/T3 phage polymerase promoter added at the 5' end of the primer. *COOLAIR* RNA was then generated by following a protocol provided by the MAXIscript® T7 Transcription Kit (Thermo Fisher Scientific). The resulting product was DNA digested, denatured, and separated on 5% TBE gel containing 8 M Urea. The band at the right size was cut out, sliced, and eluted in gel extraction buffer (0.5 M $NH_4Ac$, 1 mM EDTA, 0.2% SDS, 1xRNAsecure (Thermo Fisher Scientific) at 37 °C overnight, followed by precipitation with isopropanol. 10 pmol purified RNA was 3'biotinylated by following the instructions provided by the RNA 3' End Biotinylation Kit (Thermo Fisher Scientific). The labelling efficiency was determined to be at least 90%. For the gel shift assay, a 20 μl reaction (25 ng/μl protein, 2 nM probe, 250 ng/μl tRNA, 50 ng/μl heparin, 10 mM HEPES (pH 7.3), 120 mM KCL, 1 mM $MgCl_2$, 1 mM DTT) was set up on ice. For the competition assay, 2x to 100x of unlabelled probe was included in the reaction. The reaction with only GST was used as a negative control. The mixture was incubated at room temperature for 5 min and resolved on 3.5% native TBE-acrylamide gel and transferred onto a positively charged nylon membrane. The biotinylated RNA on the membrane was detected by chemiluminescence according to a protocol provided with the Chemiluminescent Nucleic Acid Detection Module (Thermo Fisher Scientific, 89880). For the binding reaction, 1 μg of different recombinant proteins was incubated with 2 pmol of biotinylated probe in 20 μL of 1× buffer containing 20 mM Tris (pH 7.5), 100 mM NaCl, 10 mM $MgCl_2$, non-specific competitors (1 μg/μl heparin and 5 μg tRNA) and 0.05% Triton X-100 at room temperature for 10 min. The RNA-protein mixture was resolved on 6% 1× TBE acrylamide gel under 100 V for 50 min, followed by electrophoretic transfer to positively charged nylon membranes (GE Healthcare). The biotinylated RNA on the membrane was UV cross-linked and detected using chemiluminescent nucleic acid detection following the manufacturer's instructions (Thermo Fisher Scientific).

## Data availability

Raw image data of confocal microscopy and co-localization analysis have been deposited in the BioImage Archive (https://www.ebi.ac.uk/biostudies/bioimages/studies/S-BIAD1154). The raw and processed imaging data and analysed tracks of single-particle tracking analysis are available at the BioStudies: https://doi.org/10.6019/S-BIAD1234. The ADEMScode MATLAB tracking analysis software can be found at https://github.com/alex-payne-dwyer/single-molecule-tools-alpd.

The raw reads of Quant-seq have been deposited in the Short Read Archive (SRA) under the reference (PRJNA1087342 and PRJNA1076161).

The source data of this paper are collected in the following database record: biostudies:S-SCDT-10_1038-S44318-025-00394-4.

## Peer review information

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

## Acknowledgements

We thank Shuqin Chen, Tina Zhang, JIC Horticultural Services and JIC Bioimaging facility for their excellent technical assistance and Hsuan Pai (TSL) for her artwork assistance. We thank Simon H. Stitzinger and Jose A. Morin (MPI-IE) for helpful discussion. We also thank all members of the Caroline Dean and Martin Howard groups. This work was funded by the European Research Council Advanced Grant (EPISWITCH, 833254), Wellcome Trust (210654/Z/18/Z), and the Royal Society Professorship (RSRP\R\231006) to Caroline Dean, and EPSRC grants (EP/T00214X/1 to CD and EP/T002166/1 and EP/W024063/1 to ML).

## Author contributions

**Geng-Jen Jang**: Conceptualization; Data curation; Investigation; Methodology; Writing—original draft; Writing—review and editing. **Alex L Payne-Dwyer**: Data curation; Investigation; Methodology; Writing—review and editing. **Robert Maple**: Data curation; Investigation. **Zhe Wu**: Investigation. **Fuquan Liu**: Investigation. **Sergio G Lopez**: Investigation; Methodology. **Yanning Wang**: Investigation. **Xiaofeng Fang**: Supervision; Investigation. **Mark C Leake**: Supervision; Funding acquisition. **Caroline Dean**: Supervision; Funding acquisition; Writing—review and editing.

Source data underlying figure panels in this paper may have individual authorship assigned. Where available, figure panel/source data authorship is listed in the following database record: biostudies:S-SCDT-10_1038-S44318-025-00394-4.

## Disclosure and competing interests statement

The authors declare no competing interests.

# Expanded View Figures

**Figure EV1. A missense mutation (L304F) on the second RRM of FCA attenuates RNA-binding.**

(A) Schematic representation of the transgenic FCA-mScarlet-I fusion within the FCA locus—the position of the missense L304F mutation is indicated. Untranslated regions are indicated by grey boxes, exon regions by black boxes, and mScarlet-I coding sequence by the red box. (B) Expression of *FLC* and *FCA* relative to *UBC* in plants at the T3 generation following transformation. Data represent mean values from technical repeats. The FCAwt #8 and FCArrm #1 lines are used for further analysis. (C) FCA protein structure predicted by Alphafold Protein Structure Database (AF-Q5I5A2-F1-v4, Jumper et al, 2021; Varadi et al, 2022). The arrow indicates the location of the amino acid change from the missense mutation (amino acid 213 in this protein structure database corresponds to amino acid 314 when FCA uses the non-canonical start CTG codon). (D) Peptide alignment of Arabidopsis FCA proteins from different plant species (https://phytozome-next.jgi.doe.gov). The arrow indicates the conserved Leucine 304. (E) Plants carrying transgenes with the L304F mutation show delayed flowering compared to the control. Data are mean ± s.d. (n = 23 and 24 seedlings for the control and L304F, respectively). Statistical significance was calculated using the two-tailed t-test, with *P*-value = 4.4e−23. ****P*-value < 0.0001. (F) In vitro RNA binding assay showing that the Leu 304 and each RRM domain are important.

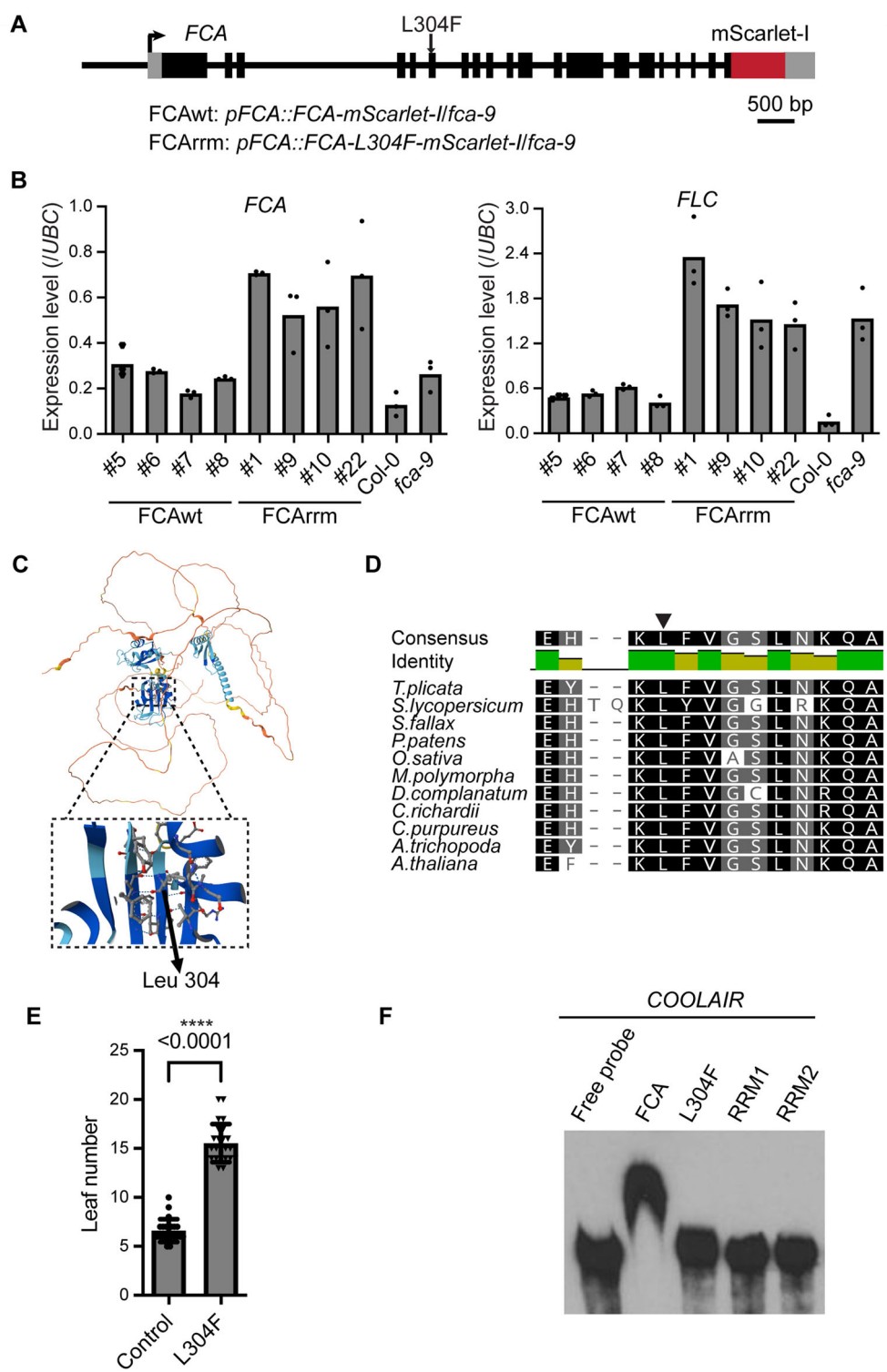

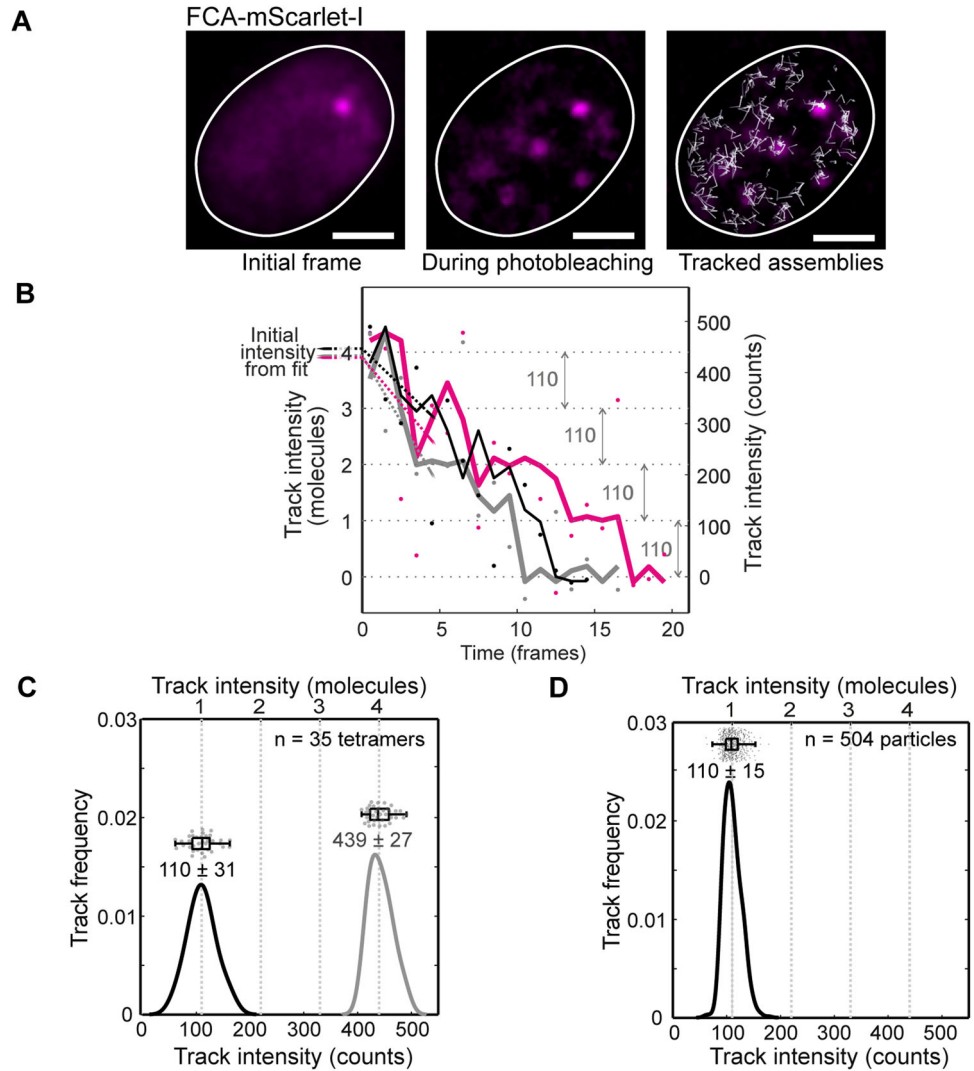

**Figure EV2. Automated tracking analysis of FCA particles and determination of the characteristic single-molecule brightness in living tissues.**

(A) An example of FCA-mScarlet-I particle tracking at 10 ms/frame. The intermediate frame, overlaid with tracks (white arrows) corresponding to the particle paths, achieves super-resolved localization precision and provides estimates of the corresponding stoichiometry and mobility. The left panel shows the initial frame of the acquisition, while the middle panels display the intermediate stage of the acquisition (after 75 ms), where contrast is greatest and assemblies are clearly visible. The nuclear boundary (white closed line) is obtained by blurring the initial frame and setting the Otsu intensity threshold. The third panel features the same detected assemblies throughout the acquisition. Scale bar: 2 μm. (B) The consistent photobleaching steps yield an estimate of about 110 photoelectron counts for the characteristic single-molecule brightness of the FCA-mScarlet-I in planta. The intensity of particles near the end of five particle tracks in the FCA-mScarlet-I image sequences is represented by square dots, and the denoised signals by solid lines of the corresponding colour. Here, zero intensity corresponds to the mean background level after complete photobleaching. (C) A subset of $n = 35$ particles was identified as having four photobleaching steps. This subset gives a characteristic single-molecule brightness (black line) of 110 ± 31 counts (mean ± s.d., $n = 35$). The initial intensities of these particles (grey line) are each greater than the characteristic single-molecule brightness by a factor of ~ four (stoichiometry: 4.0 ± 0.6 molecules, mean ± s.d., $n = 35$), suggesting that each particle is composed of ~4 FCA-mScarlet-I molecules. Data for individual particles are shown above the corresponding distribution in a jitter and box plot (showing median, interquartile range and min-max range). (D) A characteristic single-molecule brightness of 110 ± 15 counts (mean ± s.d., $n = 504$) is calculated from the time-averaged intensity levels of each of $n = 504$ particles photobleached to a single molecule above background, sifted using the signal-to-noise threshold. Data for individual particles are shown above the corresponding distribution in a jitter and box plot (showing median, interquartile range and min–max range).

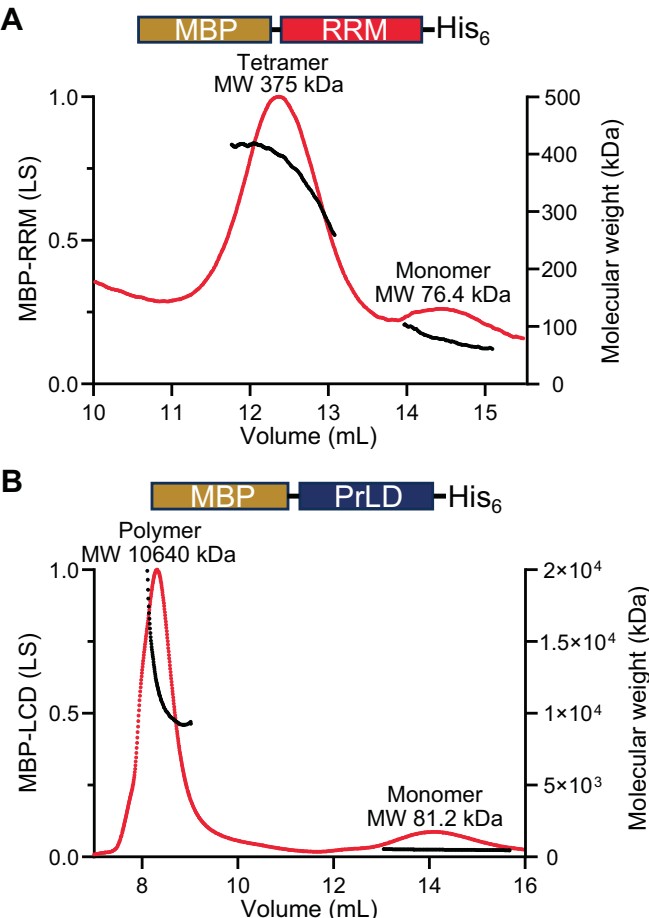

**Figure EV3. SEC-MALS of purified FCA truncated proteins.**

SEC-MALS of purified MBP-RRM (**A**) or MBP-PrLD (**B**). Line traces indicate molar mass as determined from MALS. The predicted molecular mass of MBP fused FCA truncated protein MBP-RRM and MBP-PrLD is approximately 72 kDa and 89 kDa, respectively.

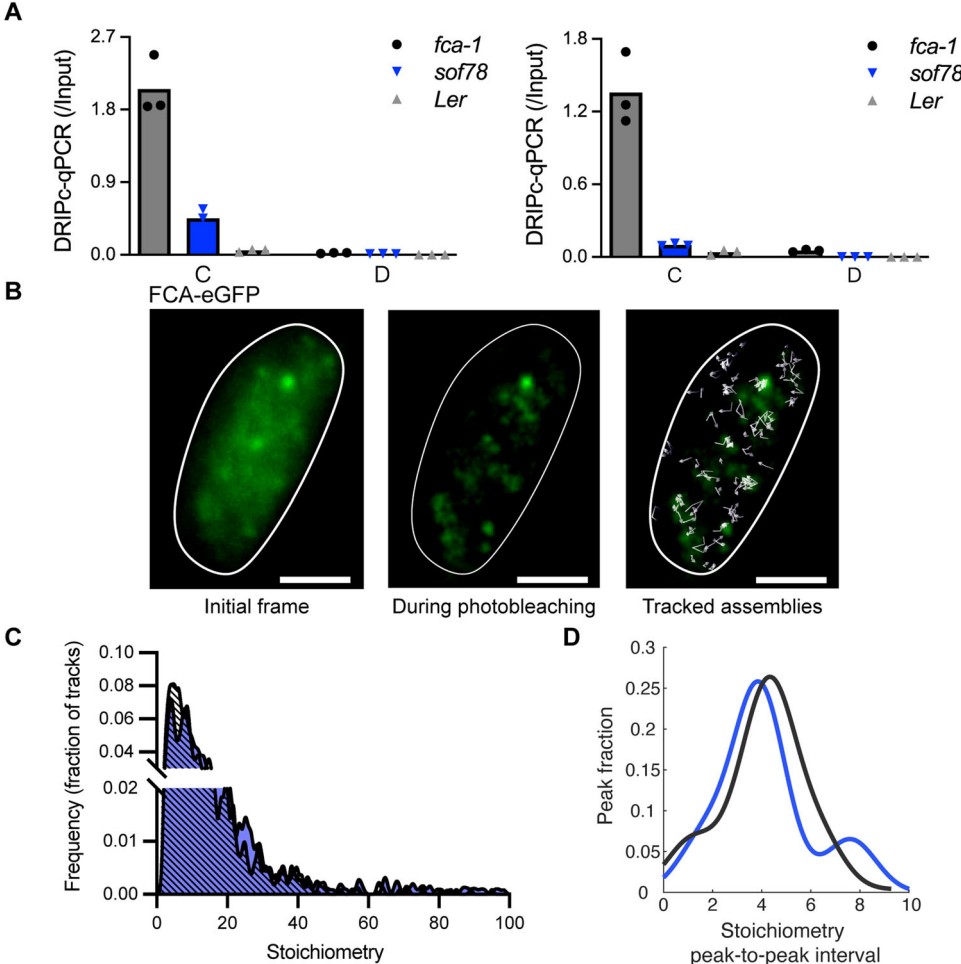

**Figure EV4. The *COOLAIR* R-loop level and FCA-eGFP single-particle tracking in the *sof78* mutant.**

(A) Two biological replicates of DRIPc–qPCR *COOLAIR* R-loop in the L*er*, *sof78* and *fca-1* genotypes. The *sof78* mutant carries a missense mutation in a region of FLL2 that is predicted to form a salt bridge, which influences FCA condensation (Fang et al, 2019). Data are means from three technical repeats. TSS, transcription start site. The primer sets used for DRIPc-qPCR (C, D) are the same as those shown in Fig. 4D. (B) An example of FCA-eGFP particle tracking at 10 ms/frame. The left panel shows the initial frame of the acquisition. The third panel features the same intermediate frame overlaid with tracks (white arrows) corresponding to the paths of each detected assembly throughout the acquisition. The nuclear boundary (white closed line) is obtained by blurring the initial frame and setting the Otsu intensity threshold. Scale bar: 2 μm. (C) Distributions of stoichiometry of individual FCA-eGFP (black) and FCA-eGFP/*sof78* (blue) particles. (D) Periodicity of FCA-eGFP (black) and FCA-eGFP/*sof78* (blue) particle stoichiometry. The stoichiometry in the figure is capped at 100. The single-particle tracking analysis is from three biologically independent experiments.

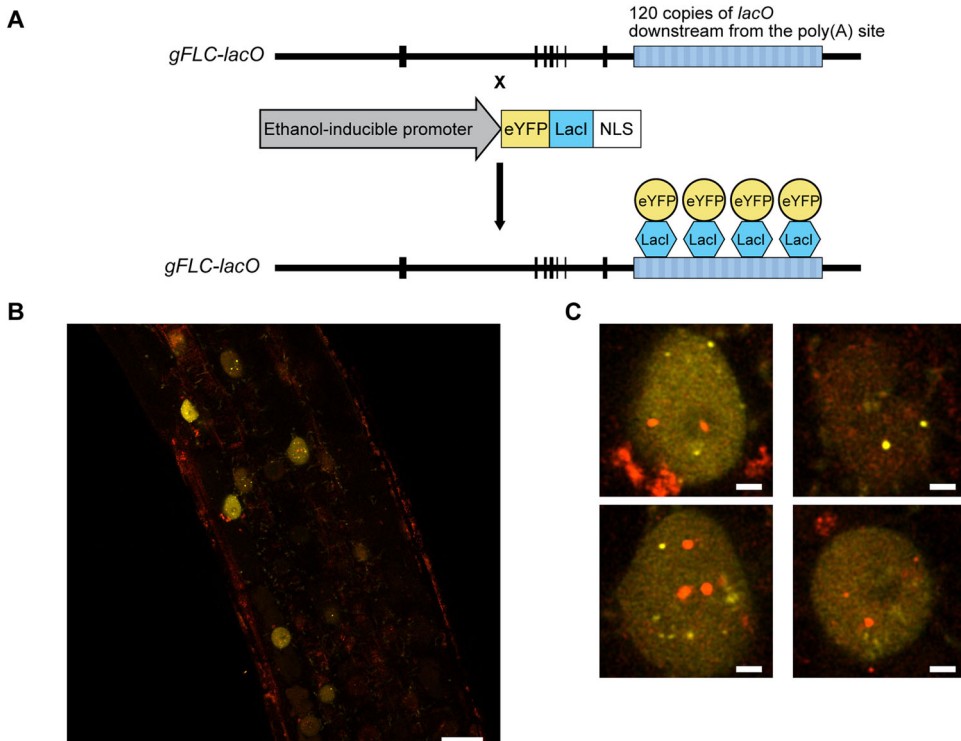

**Figure EV5. Co-localization analysis of FCA condensates and *FLC-lacO*/eYFP-LacI.**

(**A**) An illustration of the *FLC-lacO*/eYFP-LacI transgenic plants. (**B**) Single slice of Arabidopsis root Airyscan image showing expression of FCA-mScarlet-I (red) and *FLC-lacO*/eYFP-LacI (yellow) in nuclei. Scale bars: 20 μm. (**C**) Subsets of Airyscan images from (**B**) showing nuclei co-expressing FCA condensates (red) and *FLC-lacO*/eYFP-LacI (yellow). Scale bars: 2 μm. The FCA-mScarlet-I and LacI-eYFP signals were enhanced by shifting their dynamic ranges to 0–1099 and 0–4220 in EV5B, and to 0–1000 and 0–6000 in EV5C. The images were acquired on a Zeiss LSM980 using LD C-Apochromat 40x/NA 1.1 water-immersion objective. The fluorescent proteins, eYFP and mScarlet-I, were excited at 488 and 561 nm, respectively.

