## [Peer Review File · The EMBO Journal]

Modular *in vivo* assembly of *Arabidopsis* FCA oligomers into condensates competent for RNA 3' processing

Geng-Jen Jang, Alex Payne-Dwyer, Robert Maple, Zhe Wu, Fuquan Liu, Sergio Lopez, Yanning Wang, Xiaofeng Fang, Mark Leake, and Caroline Dean

Corresponding author(s): Caroline Dean (caroline.dean@jic.ac.uk)

Review Timeline:

Submission Date:	24th Jun 24
Editorial Decision:	17th Jul 24
Revision Received:	14th Dec 24
Editorial Decision:	7th Jan 25
Revision Received:	26th Jan 25
Accepted:	6th Feb 25

Editor: William Teale

Transaction Report:

Dear Dame Caroline,

Thank you again for the submission of your manuscript entitled "In vivo properties of Arabidopsis FCA condensates involved in RNA 3' processing" (EMBOJ-2024-118276). We have now received reports from two referees, which I copy below.

As you can see from their comments, while referee #1 is an enthusiastic supporter of publication, referee #2 provides an extensive list of technical concerns that I would like you to consider carefully.

Based on the overall interest expressed in the reports, however, I would like to invite you to address the comments of both referees in a revised version of the manuscript. I should add that it is The EMBO Journal policy to allow only a single major round of revision and that it is therefore important to resolve the main concerns at this stage. I believe the concerns of the referees are reasonable and addressable, but please contact me if you have any questions, need further input on the referee comments or if you anticipate any problems in addressing any of their points. I am always available for a Zoom call, should you find it helpful to discuss referee #2's comments; I think this would be particularly useful if you decide to modify any conclusions in addition to providing more methodological detail. Please, follow the instructions below when preparing your manuscript for resubmission.

I would also like to point out that as a matter of policy, competing manuscripts published during this period will not be taken into consideration in our assessment of the novelty presented by your study ("scooping" protection). We have extended this 'scooping protection policy' beyond the usual 3 month revision timeline to cover the period required for a full revision to address the essential experimental issues. Please contact me if you see a paper with related content published elsewhere to discuss the appropriate course of action.

Again, please contact me at any time during revision if you need any help or have further questions.

Thank you very much again for the opportunity to consider your work for publication. I look forward to your revision.

Best wishes,

William

William Teale, Ph.D.
Editor
The EMBO Journal

When submitting your revised manuscript, please carefully review the instructions below and include the following items:

- 1) a .docx formatted version of the manuscript text (including legends for main figures, EV figures and tables). Please make sure that the changes are highlighted to be clearly visible.
- 2) individual production quality figure files as .eps, .tif, .jpg (one file per figure).
- 3) a .docx formatted letter INCLUDING the reviewers' reports and your detailed point-by-point response to their comments. As part of the EMBO Press transparent editorial process, the point-by-point response is part of the Review Process File (RPF), which will be published alongside your paper.
- 4) a complete author checklist, which you can download from our author guidelines ([https://wol-prod-cdn.literatumonline.com/pb-assets/embo-site/Author Checklist%20-%20EMBO%20J-1561436015657.xlsx](https://wol-prod-cdn.literatumonline.com/pb-assets/embo-site/Author%20Checklist%20-%20EMBO%20J-1561436015657.xlsx)). Please insert information in the checklist that is also reflected in the manuscript. The completed author checklist will also be part of the RPF.
- 5) Please note that all corresponding authors are required to supply an ORCID ID for their name upon submission of a revised manuscript.
- 6) We require a 'Data Availability' section after the Materials and Methods. Before submitting your revision, primary datasets produced in this study need to be deposited in an appropriate public database, and the accession numbers and database listed under 'Data Availability'. Please remember to provide a reviewer password if the datasets are not yet public (see

<https://www.embopress.org/page/journal/14602075/authorguide#datadeposition>). If no data deposition in external databases is needed for this paper, please then state in this section: This study includes no data deposited in external repositories. Note that the Data Availability Section is restricted to new primary data that are part of this study.

Note - All links should resolve to a page where the data can be accessed.

8) For data quantification: please specify the name of the statistical test used to generate error bars and P values, the number (n) of independent experiments (specify technical or biological replicates) underlying each data point and the test used to calculate p-values in each figure legend. The figure legends should contain a basic description of n, P and the test applied. Graphs must include a description of the bars and the error bars (s.d., s.e.m.).

9) We would also encourage you to include the source data for figure panels that show essential data. Numerical data can be provided as individual .xls or .csv files (including a tab describing the data). For 'blots' or microscopy, uncropped images should be submitted (using a zip archive or a single pdf per main figure if multiple images need to be supplied for one panel). Additional information on source data and instruction on how to label the files are available at .

10) We replaced Supplementary Information with Expanded View (EV) Figures and Tables that are collapsible/expandable online (see examples in <https://www.embopress.org/doi/10.15252/embj.201695874>). A maximum of 5 EV Figures can be typeset. EV Figures should be cited as 'Figure EV1, Figure EV2" etc. in the text and their respective legends should be included in the main text after the legends of regular figures.

12) Our journal encourages inclusion of *data citations in the reference list* to directly cite datasets that were re-used and obtained from public databases. Data citations in the article text are distinct from normal bibliographical citations and should directly link to the database records from which the data can be accessed. In the main text, data citations are formatted as follows: "Data ref: Smith et al, 2001" or "Data ref: NCBI Sequence Read Archive PRJNA342805, 2017". In the Reference list, data citations must be labeled with "[DATASET]". A data reference must provide the database name, accession number/identifiers and a resolvable link to the landing page from which the data can be accessed at the end of the reference. Further instructions are available at .

Additional instructions for preparing your revised manuscript:

At EMBO Press we ask authors to provide source data for the main manuscript figures. Our source data coordinator will contact

you to discuss which figure panels we would need source data for and will also provide you with helpful tips on how to upload and organize the files.

We realize that it is difficult to revise to a specific deadline. In the interest of protecting the conceptual advance provided by the work, we recommend a revision within 3 months (15th Oct 2024). Please discuss the revision progress ahead of this time with the editor if you require more time to complete the revisions. Use the link below to submit your revision:

Referee #1:

The Dean group has been characterizing FCA condensates involved in RNA 3' processing and identified FLL2, which is necessary for the condensate formation and RNA processing by FCA (Fang et al 2019). In the submitted manuscript, they extended this in vivo characterization of the condensate and showed that FCA forms core tetramer that multimerise into higher order particles. FLL2 is frequently colocalized with larger FCA condensates. A missense mutation in the RNA binding domain of FCA reduced FCA condensate size but core tetramers remained.

This is a nice extension of their effort to characterize the function of RNA processing pathway by FCA condensate. The results are overall convincing and well controlled.

Still, I am not fully convinced that larger condensates are important for the FCA function. The possibility may remain that core tetramers can perform FCA function as well. FLL2 localization is enriched in larger condensates, but some proportions of the core tetramer are also colocalized with FLL2 (Fig. 2). The mutation in RRM reduced efficiency of large condensate formation but that does not necessarily mean that large condensate formation is important for the function of FCA. Loss of binding to RNA may affect FCA function and large condensate formation independently. Additional data or consideration to support the conclusion may be possible.

In any case, this is a beautiful work and will be cited frequently in future.

Referee #2:

Geng-Jen Jang et al. aimed to address a critical question concerning how the RNA-binding protein FCA regulates RNA processing through macromolecular condensation. Specifically, they sought to understand the differential functions of molecular assemblies at various oligomeric states. This inquiry stems from previous findings on FCA's oligomerization and the recognized significance of RNA-binding-mediated condensation for proper function, as established by the Dean lab and other researchers in the field. Both phenomena of RNA-binding require FAC condensation, RNA-binding is required to function in COOLAIR R-loop resolution, and FCA autoregulation is solid, respectively, and well-supported genetic materials and RNA processing assays.

The key novelty of this study aims to provide quantitative evidence on the precise assembly status of FCA oligomers and how this assembly regulates its function, such as RNA 3' processing, using in vivo imaging. The findings, interpretations, and

conclusions rely on the application of a single-particle imaging method using SlimVar microscopy, which technique has also recently been submitted to bioRxiv as another manuscript. In this manuscript, demonstrating the hypothesized quantitative stepwise assembly and function, transitioning from a low oligomeric dysfunctional complex to a functional full macromolecular assembly (condensates), could significantly advance our understanding of dynamic condensation and functional transitions. I appreciate the authors' efforts to address this important point with their complementary expertise. The RNA-RRM dependent function is well supported the importance of condensation for RNA 3' processing. However, after carefully reading the manuscript, several significant concerns regarding technical points and data interpretation prevent the reviewers from being convinced that the data provide convincing quantitative assessments of different functional assemblies to adequately support the conclusions made.

Major comments

1. The authors used in vivo imaging and analysis to conclude that the basal non-functional assembly (FCArm), a C-terminal fragment of FCA (maybe ~291-747aa) containing a prion-like domain, forms a tetramer. The author claimed that their finding aligns with biochemical results from SEC-MALS conducted in vitro on the N-terminal region (RRM domains after 291aa), which also indicated tetramer formation.

First, the reviewer finds it illogical to conclude that the full-length FCA is a tetramer (lines 104-106) based on the evidence of both a tetrameric N-terminal RRM domain and a tetrameric C-terminal prion-like domain. In fact, if both fragments are really tetramer, they are more likely suggest heterooligomeric forms in full length, rather than a definitive determination of the most stable oligomeric state, considering the nature of C-terminal as prion embedded-IDR.

Second, Figure EV3A shows that a significant portion of the RRM domain exists in high oligomeric states ranging from ~350-420 kDa, with a smaller fraction appearing tetrameric (~76 kDa, less than 10%) and another population in dimeric form. Figure EV3B illustrates that FCA-PrLD forms heterooligomeric states ranging from 10,000-18,000 kDa, suggesting it may exceed 200 mer based on the indicated molecular weight of 10,640 kDa. This stark contrast between the in vitro FCA-PrLD and the in vivo FCArm indicates potential heterogeneity in oligomeric states. While this contrast does not rule out the possibility of claimed low oligomeric states for FCArm, being shown by in vivo imaging with diffusive fluorescence signal, the SEC-MALS data from both in vitro C-terminal FCA-PrLD and N-terminal RRM domains are inappropriate to support the conclusion on line 104-106 that FCAwt forms a tetramer with "consistency" between these results.

2. The single particle tracking and stoichiometry characterization is another crucial aspect of this manuscript. The reviewer could not identify most of the original images, movies, and signal analysis quality control, and some analysis details. To understand the image setup and analysis, the reviewer was able to get some information by reading through their another submitted preprint Payne-Dwyer et al. (2024) on bioRxiv. SlimVar offers the advantage of deeper penetration and better photon collection from the nucleus, which is highly beneficial for high-contrast imaging of nuclear condensates. However, this approach also gathers more signals from Z-directions, creating challenges to achieve single-molecule imaging and single-particle analysis in crowded particle environments. Whereas stepwise bleaching on fluorescent units and bleaching corrections are well-accepted methods for investigating oligomerization on resolved particles, the key challenge for this technique in living systems is the limited spatial resolution of light microscopy and the too-high density of nanometer-sized particles. The reviewer could not understand how the SlimVar technique addresses these issues and enables a quantitative examination of single particles to test the proposed hypothesis and provide reliable quantitative results in stoichiometry within the living cell nucleoplasm.

First, regarding the resolution for single particle tracking, could the author provide the PSF measurement of the system and describe how the issue of PSF overlap can be resolved for continuous signals shown in Figure 3A FCArm and diffusive areas in Figure EV2A FCAwt? Additionally, FCAwt in Figure 1A and Figure EV2A appear markedly different concerning the density of large clusters. May authors explain the reasons behind.

Second, could the author provide details on the imaging processing and single particle tracking quality control to demonstrate how particles were selected from the smear-looking nucleoplasm FCA signals in Figure 3A (FCArm) and Figure EV2A (FCAwt)? When single particle trajectories are shown in Figure EV2, the reviewer had difficulty identifying single particles, as most of the signals are dim and diffuse, as shown in Figure EV2A. Additionally, could the author provide movies of FCA-mScarlet-I single particle dynamics to illustrate the time and spatial resolution needed to capture FCA single particles, which is the tetrameric population the author concluded?

Third, could the author kindly elaborate how the broad peak stoichiometry distribution between 0-10 on the Y-axis in Figure 1A and Figure 3A concludes a tetramer form from an imaging analysis perspective?

Fourth, the author may want to clarify what is the population of the particles that were chosen and analyzed for diffusion coefficient for FCAwt in Figure 3E. Does the statistical significance with a P-value of 0.0066 between two groups reflect the difference between large condensates and the claimed tetramer-based diffused signals? The intention of the experiment and the conclusion from the results seem to be different.

3. Accuracy and robust measurement of the signal decrease in the bleaching step and the total initial intensity of every single particle are the keys that enable the calculation of the stoichiometry of FCA in the manuscript. The single-molecule brightness is

measured as shown in Figure EV2B and divided by the track's initial intensity.

First, regarding the bleaching step, the authors describe this process: "The stoichiometry was found by extrapolating the first 5 particles' intensities to account for photobleaching." It is important to show multiple 5-step bleaching trajectories to demonstrate the quality of signal/noise for clear step-wise decrease, the accuracy in the decreased value between steps, and robustness in photo-collection with manageable frustration, as high signal variation and non-detectable bleaching are often significant among different particle analyses. Each 5-step bleaching curve could be shown next to the images of the corresponding chosen single particle. In the same line, the author can highlight the single particles from the representative images/movies of Figure 1A, Figure 3A, and Figure EV2A that were used for bleaching step measurements as well as single particle analysis. When I read the method, I guess the cited Tweezer work of Leake et al. (2003) is incorrect "The characteristic single molecule brightness linked to each reporter was determined as 110 ± 15 (mean \pm sem) for mScarlet-I based on the Chung-Kennedy-filtered terminal intensity of tracks in each acquisition (Leake et al, 2003)."

Second, the accuracy of the "initial intensity" of a single particle used to divide the value of single molecule bleaching is crucial. However, this value is often highly variable due to factors such as fluorophore properties, imaging environment, and analysis approaches, including but not limited to partial bleaching, incomplete XFP maturation, and fluctuation in Z-position at the illumination plane, as well as the limited spatial resolution to calculate the right number of particles. For example, while previous work of the author was done by TIRF (Leake et al., 2006) with a narrowed illumination plane in Z, the SlimVar has a deeper illumination in Z that would collect multiple particles in the nucleoplasm as the "initial intensity." Such uncertainty in the number of particles from a bulk signal collection would result in huge errors in defining the low-oligomeric states of a single particle. The author may elaborate on how calibrations were performed to achieve the accuracy described in the manuscript. In addition, even imaging the uniform assembled particles, the authors might have awarded that the signal intensity variation is large from decent scale of particle analysis, which need to be careful calibrated. The author may elaborate on the level and sources of signal variation and how these variations were controlled to provide reliable values for calculating stoichiometry. One reliable and practical way the reviewer could suggest to improve the reliability is to perform a careful calibration by generating a linear progression curve using low-density fluorescent particles with known and fixed oligomerization numbers in the nucleoplasm.

Minor points:

1. Regarding image analysis for colocalization, could the author clarify how the particles were chosen, as this is not evident in Figure 2A? Additionally, how were their volumes calculated in Figure 2B, given that a large portion of the values are close to zero?
2. How was the colocalization value in Figure 2D calculated for the generated plot? The reviewer can easily see around 10 large condensates with both FLL2 and FCA in Figure 2A. It seems that some images with more condensates were excluded (such as Figure 1A and 2A) from the analysis.
3. In Figure EV2B, it states "The intensity of particles near the end of four particle tracks...". Should this be five tracks instead?
4. There appears to be a typo in the sentence: "...with (FCArm) and without (FCAwt) the RRM..."

Dear EMBO Editor (William),

We thank the reviewers for their thoughtful comments. We have edited the paper, adding new experiments and revising text (all detailed below) – their words in italics, with the main points in bold – our response below each one non-italicised. As one general edit, we now carefully distinguish FCA particles that are followed in the SlimVar tracking experiments (both low-order and multimers) from FCA condensates, visualised by the confocal analysis.

Referee #1:

The Dean group has been characterizing FCA condensates involved in RNA 3' processing and identified FLL2, which is necessary for the condensate formation and RNA processing by FCA (Fang et al 2019). In the submitted manuscript, they extended this in vivo characterization of the condensate and showed that FCA forms core tetramer that multimerise into higher order particles. FLL2 is frequently colocalized with larger FCA condensates. A missense mutation in the RNA binding domain of FCA reduced FCA condensate size but core tetramers remained.

This is a nice extension of their effort to characterize the function of RNA processing pathway by FCA condensate. The results are overall convincing and well controlled.

1. Still, I am **not fully convinced that larger condensates are important for the FCA function**. The **possibility may remain that core tetramers can perform FCA function** as well. FLL2 localization is enriched in larger condensates, but some proportions of the core tetramer are also colocalized with FLL2 (Fig. 2). The mutation in RRM reduced efficiency of large condensate formation but that does not necessarily mean that large condensate formation is important for the function of FCA. Loss of binding to RNA may affect FCA function and large condensate formation independently. **Additional data or consideration to support the conclusion may be possible**.

In any case, this is a beautiful work and will be cited frequently in future.

Response: To further support the conclusion that larger condensates are important for function we have added a single particle tracking analysis of FCA-eGFP in the *sof78* mutant (Fig. EV4). We find the periodicity and stoichiometry distribution of FCA particles in *sof78* show no obvious difference compared to the control. This supports the view that it is the larger condensates, too large and uniform to be tracked as individual optical foci using SlimVar, which are functional. Thus, RNA-protein and protein-protein interactions differentially contribute to the formation of the condensates (Fig. 3 and Fig. EV4B to D).

In addition, all of our current evidence indicates that reducing size and number of the FCA condensates influence FCA's molecular function or at least decreases its efficiency in RNA 3' processing (Fig. 3A and Fig. 4C in this manuscript; Fig. 3E and Fig. 4D in Fang *et al.*, 2019). To further support this conclusion, we have now added data showing that *sof78*, a mutant predicted to disrupt a salt bridge connecting two coiled coils, attenuates *COOLAIR* R-loop resolution, like the FCArm mutant (Fig. EV4A).

Referee #2:

Geng-Jen Jang et al. aimed to address a critical question concerning how the RNA-binding protein FCA regulates RNA processing through macromolecular condensation. Specifically, they sought to understand the differential functions of molecular assemblies at various oligomeric states. This inquiry stems from previous findings on FCA's oligomerization and the recognized significance of RNA-binding-mediated condensation for proper function, as established by the Dean lab and other researchers in the field. Both phenomena of RNA-binding require FAC condensation, RNA-binding is required to function in COOLAIR R-loop resolution, and FCA autoregulation is solid, respectively, and well-supported genetic materials and RNA processing assays.

The key novelty of this study aims to provide quantitative evidence on the precise assembly status of FCA oligomers and how this assembly regulates its function, such as RNA 3' processing, using in vivo imaging. The findings, interpretations, and conclusions rely on the application of a single-particle imaging method using SlimVar microscopy, which technique has also recently been submitted to bioRxiv as another manuscript. In this manuscript, demonstrating the hypothesized quantitative stepwise assembly and function, transitioning from a low oligomeric dysfunctional complex to a functional full macromolecular assembly (condensates), could significantly advance our understanding of dynamic condensation and functional transitions. I appreciate the authors' efforts to address this important point with their complementary expertise. The RNA-RRM dependent function is well supported the importance of condensation for RNA 3' processing. However, after carefully reading the manuscript, several significant concerns regarding technical points and data interpretation prevent the reviewers from being convinced that the data provide convincing quantitative assessments of different functional assemblies to adequately support the conclusions made.

Major comments

1. The authors used in vivo imaging and analysis to conclude that the basal non-functional assembly (FCArm), a C-terminal fragment of FCA (maybe ~291-747aa) containing a prion-like domain, forms a tetramer. The author claimed that their finding aligns with biochemical results from SEC-MALS conducted in vitro on the N-terminal region (RRM domains after 291aa), which also indicated tetramer formation.

First, the reviewer finds it illogical to conclude that the full-length FCA is a tetramer (lines 104-106) based on the evidence of both a tetrameric N-terminal RRM domain and a tetrameric C-terminal prion-like domain. In fact, if both fragments are really tetramer, they are more likely suggest heterooligomeric forms in full length, rather than a definitive determination of the most stable oligomeric state, considering the nature of C-terminal as prion embedded-IDR.

Second, Figure EV3A shows that a significant portion of the RRM domain exists in high oligomeric states ranging from ~350-420 kDa, with a smaller fraction appearing tetrameric (~76 kDa, less than 10%) and another population in dimeric form. Figure EV3B illustrates that FCA-PrLD forms heterooligomeric states ranging from 10,000-18,000 kDa, suggesting it may exceed 200 mer based on the indicated molecular weight of 10,640 kDa. This stark contrast between the in vitro FCA-PrLD and the in vivo FCArm indicates potential

heterogeneity in oligomeric states. While this contrast does not rule out the possibility of claimed low oligomeric states for FCArm, being shown by *in vivo* imaging with diffusive fluorescence signal, the SEC-MALS data from both *in vitro* C-terminal FCA-PrLD and N-terminal RRM domains are inappropriate to support the conclusion on line 104-106 that FCAwt forms a tetramer with "consistency" between these results.

Response: First, we would like to clarify that the FCArm used for *in-vivo* imaging is the full-length FCA protein containing one amino acid change in the 2nd RRM domain (Fig. EV1). For the constructs used in SEC-MALS, we have modified the labelling and provided a schematic in Fig. EV3 to make it clearer. The predicted molecular mass of FCA NH2-terminal region protein, MBP-RRM (fused with the MBP solubility tag), is approximately 72 kDa. Therefore, a significant portion of the RRM region exists at a peak of 375 kDa (roughly a tetramer or possibly a pentamer), with a small fraction existing as a monomer. The predicted molecular mass of FCA C-terminal region protein, MBP-PrLD (fused with the MBP solubility tag), is approximately 89 kDa. Figure EV3B shows that MBP-PrLD forms megacomplexes of more than 100 mers; likely reflecting *in vitro* aggregation of PrLDs when expressed at high concentrations *in vitro*. In the revised version, we have clearly labelled the size of each peak to reduce confusion.

*The single particle tracking and stoichiometry characterization is another crucial aspect of this manuscript. The reviewer could not **identify most of the original images, movies, and signal analysis quality control, and some analysis details**. To understand the image setup and analysis, the reviewer was able to get some information by reading through their another submitted preprint Payne-Dwyer et al. (2024) on bioRxiv. SlimVar offers the advantage of deeper penetration and better photon collection from the nucleus, which is highly beneficial for high-contrast imaging of nuclear condensates. However, this approach also gathers more signals from Z-directions, creating challenges to achieve single-molecule imaging and single-particle analysis in crowded particle environments. Whereas stepwise bleaching on fluorescent units and bleaching corrections are well-accepted methods for investigating oligomerization on resolved particles, the key challenge for this technique in living systems is the **limited spatial resolution** of light microscopy and the **too-high density** of nanometer-sized particles. The reviewer **could not understand how the SlimVar technique addresses these issues** and enables a quantitative examination of single particles to test the proposed hypothesis and provide reliable quantitative results in stoichiometry within the living cell nucleoplasm.*

*First, regarding the resolution for single particle tracking, **could the author provide the PSF measurement of the system***

Response: SlimVar combines the tracking capabilities of Slimfield with significant improvements in imaging contrast from variable-angle laser excitation. Our use of an oil objective lens is a compromise that improves the photon collection efficiency but introduces optical aberrations that blur the point spread function (PSF) at high imaging depths, especially in the z-direction as the reviewer suggests. However, we use optical hardware adjustments such as the objective lens' correction collar to minimise these aberrations at a specified calibration depth of 20 microns. These optical improvements are demonstrated by 3D PSF estimates both *in vitro* and in root tips (Figure attached below, the result will be published in the technical section of Payne-Dwyer et al., 2024a. This calibration strategy is

commonly used in optical tweezers experimentation to increase laser trap stiffness (a proxy for PSF contrast) by threefold at depths of up to 50 microns in water (Dienerowitz *et al.*, 2011).

While the calibration does not fully recover diffraction-limited imaging at depth, it does so to a level acceptable for our downstream postprocessing. We have added a paragraph in the discussion section to address the limitations of our approach and the current methodology.

*and **describe how the issue of PSF overlap can be resolved for continuous signals** shown in Figure 3A FCArm and diffusive areas in Figure EV2A FCAwt?*

Response: As the reviewer suggests, the overlap limits depend on the PSF dimensions and the number of assemblies. At high density and mobility, overlapping or blurred signals contribute to the background, reducing the contrast of in-focus objects. This loss of contrast would normally lower the apparent signal intensities and distort the stoichiometry estimates.

While our experimental PSF does not achieve diffraction-limited performance, it both facilitates and benefits from postprocessing “sifting” in which noisy signals become robust to

moderate changes in PSF shape, for example due to defocus, motion blur, and residual optical aberrations (see also comment 3.2. part 1). The sifting uses a tracking acceptance criterion that the measured levels of the summed signal and local background around each focus must meet a predefined signal-to-noise threshold. Not only does this eliminate false positive foci due to noise, but rescues foci with lowered peak contrast due to blur, and discards foci with insufficient total contrast. Fluorescent foci whose intensity profiles extend beyond approximately three times the diffraction-limited PSF full width at half maximum (FWHM), or which overlap too closely with those of neighbouring foci, are not included in subsequent tracking quantification. Such overlapping or blurry particles instead contribute to the global fluorescence background and do not bias the calculated signal intensities of the remaining tracked foci, which remain representative of the population. This combined approach of contrast enhancement and postprocessing sifting results in an increase in measured foci signal-to-noise ratio values at the 20-micron calibration depth by a factor of approximately 3, also enabling single particle tracking to depths up to 30 microns inside plant root tissue. We have modified the Method section to make this explanation clearer.

*Additionally, FCAWT in Figure 1A and Figure EV2A appear **markedly different concerning the density of large clusters. May authors explain the reasons** behind.*

Response: The purpose of Fig. 1A is to describe SlimVar microscopy for FCA particle tracking. Therefore, we selected an image with multiple particles to demonstrate the starting material. In EV2A, we demonstrate how particle tracking works. We show the acquisitions as their initial frames without matched contrast and provide movies for these two figures (Movie EV1 and EV2), which – except for the large FCA particle - are more similar in the density of FCA particles than they first appear.

*Second, could the author **provide details on the imaging processing and single particle tracking quality control to demonstrate how particles were selected** from the smear-looking nucleoplasm FCA signals in Figure 3A (FCArm) and Figure EV2A (FCAwt)?*

Response: The tracks are selected from the background according to the automated quality control step (described above).

*When single particle trajectories are shown in Figure EV2, the reviewer had **difficulty identifying single particles, as most of the signals are dim and diffuse**, as shown in Figure EV2A.*

Response: We originally only showed the initial frame of the images, which did not display the maximum contrast. We have now added the intermediate stage of the acquisition, where assemblies are clearly visible, and provided movies (Movie EV1 and EV2).

*Additionally, could the author **provide movies of FCA-mScarlet-I single particle dynamics to illustrate the time and spatial resolution needed to capture FCA single particles, which is the tetrameric population** the author concluded?*

Response: For direct, internally calibrated detection of molecular assemblies labelled with fluorescent fusions, we use SlimVar with rapid exposure (2-20 ms/frame) and near-diffraction-limited widefield resolution of ~180 nm and published the result in Payne-Dwyer *et al.*, 2024a. To illustrate the resolution required to detect FCA-mScarlet-I particles with

sufficient contrast, we simulated the effect of different exposure times (0.1-1000 ms/frame) using a real SlimVar example obtained at 10 ms/frame. We find this frame rate provides optimal contrast for detection of FCA-eGFP and FCA-mScarlet-I particles at the ~180 nm spatial resolution. We add a statement in the method section and summarise this in an Appendix Figure S6.

Third, could the author kindly elaborate how the broad peak stoichiometry distribution between 0-10 on the Y-axis in Figure 1A and Figure 3A concludes a tetramer form from an imaging analysis perspective?

Response: Starting with the set of floating-point stoichiometry estimates for each track, we collated and visualized the distributions in Figs. 1 and 3. For simplicity, we used a representative value of 0.6 molecules as a fixed kernel width in visualizing the stoichiometry distributions. As a result, the low-end stoichiometry peaks have a conservative uncertainty and appear blurred in this representation. The modal value remains at ~4, indicating that tetramers dominate over other constituent species such as dimers or monomers, while smaller peaks or shoulders are evident at approximate multiples of 4. The square-root error scaling is accounted for in the periodicity analysis across all the floating-point stoichiometry estimates, which confirms a tetrameric structure. For further validation, we have independently identified examples of oligomeric particles and counted four consecutive photobleaching steps indicating a stoichiometry of ~4 (Fig. EV2B-D).

Fourth, the author may want to clarify what is the population of the particles that were chosen and analyzed for diffusion coefficient for FCAwt in Figure 3E. Does the statistical significance with a P-value of 0.0066 between two groups reflect the difference between large condensates and the claimed tetramer-based diffused signals? The intention of the experiment and the conclusion from the results seem to be different.

Response: The population of tracks shown in Fig. 3E is the same as described in the stoichiometry analysis and is not subject to additional selection. This population includes all particles detectable by SlimVar above the SNR threshold and with a FWHM below the 5-pixel cutoff, which would exclude very large condensates. In our Fig. 3D, we also we compared the diffusion coefficients of individual particles with their stoichiometry.

Accuracy and robust measurement of the signal decrease in the bleaching step and the total initial intensity of every single particle are the keys that enable the calculation of the stoichiometry of FCA in the manuscript. The single-molecule brightness is measured as shown in Figure EV2B and divided by the track's initial intensity.

*First, regarding the bleaching step, the authors describe this process: "**The stoichiometry was found by extrapolating the first 5 particles' intensities to account for photobleaching.**" It is important to show multiple 5-step bleaching trajectories to demonstrate the quality of signal/noise for clear step-wise decrease, the accuracy in the decreased value between steps, and robustness in photo-collection with manageable frustration, as high signal variation and non-detectable bleaching are often significant among different particle analyses. Each 5-step bleaching curve could be shown next to the images of the corresponding chosen single particle. In the same line, the author can highlight the*

single particles from the representative images/movies of Figure 1A, Figure 3A, and Figure EV2A that were used for bleaching step measurements as well as single particle analysis.

Response: The reviewer correctly identified that the sentence was misleading. We have corrected it to the 'first 5 image frames' for each track or multimolecular assembly – the intensities in these frames are used to extrapolate back to an initial time point to estimate the unbleached intensity, independent of other tracks. We indicate this visually in a revised Fig. EV2B.

When I read the method, I guess the cited Tweezer work of Leake et al. (2003) is incorrect "The characteristic single molecule brightness linked to each reporter was determined as 110 ± 15 (mean \pm sem) for mScarlet-I based on the Chung-Kennedy-filtered terminal intensity of tracks in each acquisition (Leake et al, 2003)."

Response: The way we had placed this citation was misleading; it was intended to support our choice of the Chung-Kennedy denoising filter, rather than the tweezing use case or the characteristic single molecule brightness itself. We have now changed the citation to refer directly to the original work in which the filter is described (Chung and Kennedy, 1991).

*Second, the accuracy of the "initial intensity" of a single particle used to divide the value of single molecule bleaching is crucial. However, this value is often highly variable due to factors such as fluorophore properties, imaging environment, and analysis approaches, including but not limited to partial bleaching, incomplete XFP maturation, and fluctuation in Z-position at the illumination plane, as well as the limited spatial resolution to calculate the right number of particles. For example, while previous work of the author was done by TIRF (Leake et al., 2006) with a narrowed illumination plane in Z, the SlimVar has a deeper illumination in Z that would collect multiple particles in the nucleoplasm as the "initial intensity." Such uncertainty in the number of particles from a bulk signal collection would result in huge errors in defining the low-oligomeric states of a single particle. **The author may elaborate on how calibrations were performed to achieve the accuracy described in the manuscript. In addition, even imaging the uniform assembled particles, the authors might have awarded that the signal intensity variation is large from decent scale of particle analysis, which need to be careful calibrated. The author may elaborate on the level and sources of signal variation and how these variations were controlled to provide reliable values for calculating stoichiometry.** One reliable and practical way the reviewer could suggest to improve the reliability is to perform a careful **calibration** by generating a linear progression curve using low-density fluorescent particles **with known and fixed oligomerization numbers in the nucleoplasm.***

Response: SlimVar employs a sifting approach similar to existing Slimfield methodology, using verified heuristics and assumptions suitable for its operating time and length scales. This method ensures two key features: a low coefficient of variance in detected intensities for stoichiometries in the observed range, and the exclusion of foci and tracks that do not maintain a linear intensity relationship, regardless of the illuminated volume or background level.

First, we address how SlimVar accounts for the sources of signal variation in the apparent number and initial intensity of particles (see also above). The axial fluctuation of emitters presents similarly to lateral motion blur. Our detection algorithm uses an

integrated radius of 5 pixels (~250 nm) around each foci's centroid, slightly larger than the diffraction limit allows for a limited amount of defocus and/or motion blur, and ensures that the integrated energy (above the local background near this area) remains approximately conserved for a specific emitter. During sifting, foci beyond this size are discarded because they no longer meet the signal-to-noise ratio (SNR) condition for sufficient total contrast. Signals contributing to the average background do not affect the initial intensity of a track, but reduce its detectability.

The initial intensity of a track is estimated by back-extrapolating the integrated energy of foci to a point prior to photobleaching. We use a linear fit over the first five frames of a track to provide an estimate robust to fluctuations in the initial frame. Even the apparently high particle density in the initial frame, the relative chance of individual assemblies consistently overlapping over five subsequently photobleached frames is very low (<5%, Payne-Dwyer *et al.*, 2024b).

Spatial resolution versus emitter density does not limit single molecule brightness determination, since we focus on the photobleached end of the acquisition where only single-molecule events are visible. The photobleaching of individual fluorophores is uncorrelated, making single molecule brightness independent of stepwise photobleaching. The optical geometry ensures equal illumination of all fluorophores in the detection volume.

We discussed the nonzero possibility that multiple (uncorrelated) assemblies are captured within a single focus or track. Even were this to occur, the effect is to sum the underlying stoichiometries, and thus these rare events disproportionately affect the upper (tail) end rather than the lower end of the stoichiometry distribution.

Second, we address the influence on the emission intensity associated with each fluorophore. Equivalently, this is the average height of each photobleaching step, which we call the *characteristic single molecule brightness*. For the SlimVar experiments in this study, we used fluorescent proteins, whose average environment remained consistent across the experiments' spatial and temporal scales. For example, photophysical blinking occurs on much shorter timescales (<0.1 ms) than the frame exposure and is averaged out, while the pH is effectively constant for all fluorophores in the nucleoplasm. Unlike dyes, fluorescent proteins have negligible density-dependent self-quenching. Depth-associated scattering and aberrations also reduce the effective single molecule brightness from the value in vitro, but these occur almost uniformly within the field of view and across the range of cells we investigated. As a result, a consistent single brightness value, within 14% variability, characterises both each fluorophore and the population in a given replicate set.

In our stoichiometric analysis of particles, precise counting requires uniform labelling of proteins of interest. We assume near-complete maturation based on literature (e.g., mScarlet-I exceeds 90% maturation within 90 minutes (Guerra *et al.*, 2022)). We use endogenous 1-1 labelling and treat the sample carefully without pre-exposure, such that all FCA molecules are labelled, initially primed for fluorescence and can only be irreversibly photobleached in single independent events during the acquisition.

Third, this control delivers quantitative stoichiometry for all oligomers below 50. Based on the single molecule standard error on intensity of 14%, and scaling of shot-noise limited uncertainty in proportion with square root stoichiometry, one may confidently assign

integer stoichiometry of assemblies containing up to $(1/0.14)^2 \sim 50$ monomers, even though the frame-to-frame intensity variation of an individual molecule is closer to 60% (0.6 molecules), which we use conveniently for our kernel density estimates. For stoichiometry greater than 50, the mean value is accurate but the uncertainty is greater than one molecule of FCA.

Fourth, oligomeric controls maybe helpful but not be strictly necessary for this study.

Photoblinking of individual fluorophores suffices for accurate single molecule counting up to a stoichiometry of 50 as described above. Oligomeric controls, while challenging in plants, have many examples both *in vitro* and *in vivo* Slimfield literature. In addition, SlimVar has a precedent in detecting eYFP-LacI tetramers, which have a known oligomeric structure, *in vivo* at depth inside Arabidopsis root tips (Supplementary Fig. 7, Payne-Dwyer *et al.*, 2024a).

We added a paragraph in the discussion section to discuss limitations of current methodology in this study.

Minor points:

M1. Regarding image analysis for colocalization, could the author clarify how the particles were chosen, as this is not evident in Figure 2A? Additionally, how were their volumes calculated in Figure 2B, given that a large portion of the values are close to zero?

Response: We used the blob finder in Arivis Vision4D ver. 4.1.0. (Zeiss), which can perform 3D segmentation close to sphere-like shapes in a noisy image. We did the segmentation with specific thresholds for diameter values, probability, and split sensitivity for each protein and calculated the volumes of the identified foci (see Methods). We previously calculated the volume of overall FCA, overall FLL2, and the overlapping regions between FCA and FLL2 identified foci, and then calculated the frequency distribution. This is the possible reason that a large portion of the values is close to zero. Now, for co-localization analysis, we have shifted to calculating the volume of the FCA-identified foci that are co-localized with FLL2 (Fig. 2 B and C), which provides clearer insight into how FCA and FLL2 predominantly co-localize in the larger FCA condensates. We also lowered the probability threshold to 37% and increased the threshold for the minimum size to $0.03 \mu\text{m}^3$ for foci identification. We added one more replicate imaged from a different confocal microscope (LSM980) and observed similar trends (Appendix Fig. S2).

How was the colocalization value in Figure 2D calculated for the generated plot? The reviewer can easily see around 10 large condensates with both FLL2 and FCA in Figure 2A. It seems that some images with more condensates were excluded (such as Figure 1A and 2A) from the analysis.

Response: We previously used strength probability threshold to identify the evident condensates. As the response for M1, we have lowered the probability threshold and observed more co-localizations. This adjustment still leads to the same conclusion: FCA and FLL2 have a higher probability of co-localizing in larger condensates. We have added an Appendix Fig. S1 to show the front view and side view of the 3D images with grid lines. However, we cannot exclude the possibility that we underestimate the size and number of condensates during the co-localization analysis due to the limitations of the microscopes and

algorithm. We also edited the description of the co-localization results to make it more precise.

In Figure EV2B, it states "The intensity of particles near the end of four particle tracks...". Should this be five tracks instead?

Response: Yes. we have edited the figure legend and correct this to five.

There appears to be a typo in the sentence: "...with (FCArm) and without (FCAwt) the RRM..."

Response: We apologize for the confusion. As we clarified for major point 1, the FCArm used for in-vivo imaging contains an amino acid change in the 2nd RRM domain of full-length FCA proteins. We have edited the sentence to avoid the confusion.

References

Dienerowitz M, Gibson G, Bowman R & Padgett M (2011) Holographic aberration correction: optimising the stiffness of an optical trap deep in the sample. *Opt Express* 19: 24589

Payne-Dwyer AL & Leake MC (2022) Single-molecular quantification of flowering control proteins within nuclear condensates in live whole *Arabidopsis* root. *Methods Mol Biol.* 2476:311–328

Payne-Dwyer AL, Jang GJ, Dean C & Leake MC (2024a) SlimVar : rapid in vivo single-molecule tracking of chromatin regulators in plants. *bioRxiv* 2024.05.17.594710

Payne-Dwyer A, Kumar G, Barrett J, Gherman LK, Hodgkinson M, Plevin M, Mackinder L, Leake MC & Schaefer C (2024b) Predicting Rubisco-linker condensation from titration in the dilute phase. *Phys Rev Lett* 132: 218401

Guerra P, Vuilleminot L-A, Rae B, Ladyhina V & Miliias-Argeitis A (2022) Systematic in vivo characterization of fluorescent protein maturation in budding yeast. *ACS Synth Biol* 11: 1129–1141

Resource Data

Raw image data of confocal microscopy and co-localization analysis have been deposited in the BioImage Archive (<https://www.ebi.ac.uk/biostudies/bioimages/studies/S-BIAD1154>).
<https://www.ebi.ac.uk/biostudies/bioimages/studies/S-BIAD1154?key=e6c66502-aab8-42ee-8214-37fb890c5c26>

Raw image data of SlimVar microscopy have been deposited. (<https://doi.org/10.6019/S-BIAD1217>.)
<https://www.ebi.ac.uk/biostudies/bioimages/studies/S-BIAD1234?key=ef1a99e9-f973-4352-8f6f-b32b99f5b4c5>

The raw reads of Quant-seq have been deposited in the Short Read Archive (SRA) under the reference (PRJNA1087342 and PRJNA1076161).

<https://dataview.ncbi.nlm.nih.gov/object/PRJNA1087342?reviewer=avqm4qcb4imqkqg3npqi0965hs>

PRJNA1076161 is published.

Dear Caroline,

We have now received re-review reports from referee #2, which I have included below. As you will see, all concerns have now been addressed satisfactorily; however, I would like you to consider discussing their remaining points. Before I can finally accept the manuscript, there are some remaining editorial points which need to be addressed. In this regard would you please:

- include a callout in the manuscript text for Fig. 5A-B,
- include a title page that reads "Appendix for 'In vivo properties of Arabidopsis FCA condensates involved in RNA 3' processing'" and a table of contents with page numbers of the listed items,
- save Source data files in a scheme of one figure/folder and then upload as .zip files. E.g. all the Source data files for figure 1 need to be saved in a single folder and this needs to be zipped and then uploaded as "SD figure 1.zip" file. For EV and/or appendix figures, ZIP together all source data,
- provide exact p values in the legends of figures 2B, D; 3E, 4A, C; and EV1 E,
- define box plots in terms of minima, maxima, centre, bounds of box and whiskers, and percentile in the legend of figure EV4 D,
- define n is missing in the legend of figures 3E,
- describe the nature of entity for 'n' in the legends of figures 4A, B, C; 5B,
- remove Table EV1 from the manuscript file as it is already uploaded as an individual file, and
- zip the movie legends with each movie file.

We include a synopsis of the paper (see <http://emboj.embopress.org/>). Please provide me with a general summary image, a two sentence statement and 3-5 bullet points that capture the key findings of the paper.

I am looking forward to receiving your revised manuscript.

EMBO Press is an editorially independent publishing platform for the development of EMBO scientific publications.

Best wishes,

William

William Teale, PhD
Editor
The EMBO Journal
w.teale@embojournal.org

Please remember: Digital image enhancement is acceptable practice, as long as it accurately represents the original data and conforms to community standards. If a figure has been subjected to significant electronic manipulation, this must be noted in the

figure legend or in the 'Materials and Methods' section. The editors reserve the right to request original versions of figures and the original images that were used to assemble the figure.

We realize that it is difficult to revise to a specific deadline. In the interest of protecting the conceptual advance provided by the work, we recommend a revision within 3 months (7th Apr 2025). Please discuss the revision progress ahead of this time with the editor if you require more time to complete the revisions. Use the link below to submit your revision:

Referee #2:

I appreciate the authors' thorough and detailed responses to my previous comments. After carefully reviewing the revised manuscript and their clarifications in the rebuttal, I believe the authors have adequately addressed my concerns regarding imaging techniques and data analysis. The manuscript has been significantly improved, with the authors providing additional clarity on the SlimVar technique, PSF measurement, particle selection quality control, and bleaching-based stoichiometry measurements. They have also clarified the resolution limitations due to optical aberrations and the use of a correction collar, as well as providing more supporting data and movies to address discrepancies in the previous version.

One minor suggestion relates to the SEC-MALS analysis of FCA fragments. While the authors have clarified their methodology, my original concern was the "inconsistency" between in vitro and in vivo data regarding the tetrameric and oligomeric states of two respective domains. If one domain is a tetramer in vitro and the other domain forms oligomers, the full length will be high-oligomer. This would not fully support the tetrameric state of the full-length FCA in vivo. To address this, I suggest the authors add better discussion on why the RRM region decides the basal tetrameric state and how the PrLD or other interactions do not trigger larger assemblies unless needed. Without this clarification, the in vitro and in vivo data may seem inconsistent. Actually, SlimVar can test whether PrLD is a monomer or oligomer in vivo, but I believe a thoughtful discussion is sufficient at this stage without affecting the main conclusions.

Overall, the authors have effectively addressed the major concerns raised in the previous review, and the revisions have clarified critical points in the methodology, data interpretation, and image processing. I now feel confident in the reliability of their findings and therefore recommend an acceptance of the manuscript when a better discussion is added.

Referee #2:

I appreciate the authors' thorough and detailed responses to my previous comments. After carefully reviewing the revised manuscript and their clarifications in the rebuttal, I believe the authors have adequately addressed my concerns regarding imaging techniques and data analysis. The manuscript has been significantly improved, with the authors providing additional clarity on the SlimVar technique, PSF measurement, particle selection quality control, and bleaching-based stoichiometry measurements. They have also clarified the resolution limitations due to optical aberrations and the use of a correction collar, as well as providing more supporting data and movies to address discrepancies in the previous version.

*One minor suggestion relates to the SEC-MALS analysis of FCA fragments. While the authors have clarified their methodology, my original concern was the "inconsistency" between *in vitro* and *in vivo* data regarding the tetrameric and oligomeric states of two respective domains. If one domain is a tetramer *in vitro* and the other domain forms oligomers, the full length will be high-oligomer. This would not fully support the tetrameric state of the full-length FCA *in vivo*. To address this, I suggest the authors add better discussion on why the RRM region decides the basal tetrameric state and how the PrLD or other interactions do not trigger larger assemblies unless needed. Without this clarification, the *in vitro* and *in vivo* data may seem inconsistent. Actually, SlimVar can test whether PrLD is a monomer or oligomer *in vivo*, but I believe a thoughtful discussion is sufficient at this stage without affecting the main conclusions.*

Overall, the authors have effectively addressed the major concerns raised in the previous review, and the revisions have clarified critical points in the methodology, data interpretation, and image processing. I now feel confident in the reliability of their findings and therefore recommend an acceptance of the manuscript when a better discussion is added.

Response: Thank you, we hope our editing of the discussion (starting around line 218) to address why the RRM region may decide the basal *in vivo* tetrameric state clarifies this important point.

Dear Caroline,

I am pleased to inform you that your manuscript has been accepted for publication in the EMBO Journal.

Congratulations to all involved!

Best wishes,

William

William Teale, PhD
Editor
The EMBO Journal
w.teale@embojournal.org
